# The neurons that mistook a hat for a face

**Michael J Arcaro[1]\*, Carlos Ponce[2], Margaret Livingstone[3]**

[1]Department of Psychology, University of Pennsylvania, Philadelphia, United States; [2]Department of Neuroscience, Washington University in St. Louis, St. Louis, United States; [3]Department of Neurobiology, Harvard Medical School, Boston, United States

**Abstract** Despite evidence that context promotes the visual recognition of objects, decades of research have led to the pervasive notion that the object processing pathway in primate cortex consists of multiple areas that each process the intrinsic features of a few particular categories (e.g. faces, bodies, hands, objects, and scenes). Here we report that such category-selective neurons do not in fact code individual categories in isolation but are also sensitive to object relationships that reflect statistical regularities of the experienced environment. We show by direct neuronal recording that face-selective neurons respond not just to an image of a face, but also to parts of an image where contextual cues—for example a body—indicate a face ought to be, even if what is there is not a face.

**\*For correspondence:**
marcaro@sas.upenn.edu

**Competing interests:** The authors declare that no competing interests exist.

## Introduction

Our experience with the visual world guides how we understand it. The regularity of our experience of objects within environments and with each other provides a rich context that influences our perception and recognition of objects and categories (*Bar, 2004*; *Greene, 2013*). We do not often see faces in isolation, rather our experience of faces is embedded within a contextually rich environment – a face is usually conjoined to a body – and this regular structure influences how we perceive and interact with faces (*Bar and Ullman, 1996*; *Meeren et al., 2005*; *Rice et al., 2013*). However, it is generally found that neurons critical for the perception of faces (*Afraz et al., 2015*; *Afraz et al., 2006*; *Moeller et al., 2017*) do not respond to bodies, and vice versa (*Freiwald and Tsao, 2010*; *Freiwald et al., 2009*; *Kiani et al., 2007*; *Tsao et al., 2006*), suggesting that such neurons code for the intrinsic features of either faces or bodies but not both (*Chang and Tsao, 2017*; *Issa and DiCarlo, 2012*; *Leopold et al., 2006*). These studies typically probe neural function by presenting objects in isolation, independent of their typical context.

Here, we reconcile this apparent discrepancy between behavior and neural coding. We recorded from two face-selective domains within inferotemporal cortex (IT) in two macaque monkeys while they viewed complex natural images. By considering the spatial relationships of objects in a natural scene relative to a neuron's receptive field, we show that face-selective IT neurons respond not just to an image of a face, but to parts of an image where contextual cues—such as a body—indicate a face ought to be, even if what is there is not a face. Thus, we find that face cells demonstrate a large contextual surround tuned to entities that normally co-occur with faces. These results reveal surprisingly complex and heterogeneous inputs to face-selective neurons, suggesting that IT neurons do not represent objects in isolation, but are also sensitive to object relationships that reflect statistical regularities of the experienced environment.

## Results

We recorded simultaneously from face-selective neurons in the fMRI-defined middle lateral (ML) and posterior lateral (PL) face patches in one rhesus macaque monkey, Monkey 1, and from ML in

another monkey, Monkey 2, using chronically implanted 32-channel multi-electrode arrays (*Figure 1—figure supplement 1*). When mapped using ~2°x2° faces and round non-face objects, the activating regions were small (1-3°), located in the central visual field (within 2° of fixation, and largely contralateral; *Figure 1—figure supplement 2*). When tested with images of faces, hands, bodies (only for ML arrays), and objects presented centered on this activating region, all the visually responsive sites in both monkeys responded more to faces than to any other category (*Figure 1—figure supplement 3*). Category selectivity was assessed with an index (category1-response –category2-response)/ (category1-response + category2-response). In particular, in both monkeys' ML face patches, responses to faces were much larger than to non-face images (mean channel face vs. non-face selectivity indices = 0.63 +/- 0.06 STD and 0.64 +/- 0. 14 for Monkey's 1 and 2) or to bodies (mean channel face vs. body selectivity indices = 0.77 +/- 0.11 and 0.60 +/- 0.21). Responses to bodies were weak (body vs. object selectivity indices = −0.54 +/- 0.17 STD and 0.04 +/- 0.27 for ML in both Monkeys 1 and 2). Such a strong preference for faces over other categories is consistent with many previous studies (*Bell et al., 2011*; *Tsao et al., 2006*). The array in Monkey 1's PL face patch was less face selective (mean channel face vs. non-face selectivity indices = 0.24 +/- 0.07 STD) and responses to hands were weak (mean channel hands vs. objects indices = 0.04 +/- 0.02 STD), also consistent with previous work (*Issa and DiCarlo, 2012*).

To explore how response properties of face neurons represent naturalistic scenes, we mapped the spatial organization of face-selective neurons by presenting complex, familiar and unfamiliar scenes at various positions relative to each site's activating region (*Figure 1*). Conventionally IT neurons are studied by presenting single-object images at a single position while the animal holds fixation (*Figure 1—figure supplement 3*). Here instead, while the animal held fixation, we presented large 16° x 16° complex natural images at a series of different positions, such that different parts of the image were centered over the receptive field across presentations (*Figure 1*), in order to resolve the discrete spatial pattern of responses for each site to different parts of the image. The recording site illustrated in *Figure 1C* responded most strongly to the parts of that image that contained monkey faces. We had 28 to 32 visually responsive sites in each of the three implanted arrays. Within each array the responses were all face selective (*Figure 1—figure supplement 3*) and showed similar receptive-field locations and spatial response patterns to complex scenes (*Figure 1—figure supplements 2* and *4*). Given the uniformity of responses across channels in each array, we combined all visually responsive sites in each array for all subsequent analyses to improve signal to noise and present population-level results. Importantly, our findings are not contingent on individual unit/recording site effects. *Figure 1* shows the spatial pattern of responses to a series of images from the PL array in Monkey 1, and from the ML arrays in Monkeys 1 and 2. Consistent with the PSTHs of image categories presented in isolation (*Figure 1—figure supplement 3*), all three arrays showed responsiveness that was spatially specific to the faces in each image. Consistent with prior physiological and anatomical work suggesting that face patches are organized into a hierarchy that varies along the posterior-anterior axis (*Grimaldi et al., 2016*; *Moeller et al., 2008*), ML recording sites were more face selective and had longer latencies compared to the PL sites during simultaneous recordings (*Figure 1—figure supplement 5*). Activity specific to the eyes is apparent in both PL and ML (*Figure 2*). Activity encompassing the rest of the face emerges at longer latencies. This is consistent with prior work showing a preference for the contralateral eye in area PL that emerges between 60–80 ms post stimulus onset and longer latencies for images without a contralateral eye (*Issa and DiCarlo, 2012*). A similar shift in response latencies was evident when replacing the eyes with a visible occluder, the background of the image, and surrounding texture of the face (*Figure 2—figure supplement 1*). In our recordings, both PL and ML exhibited a shift in the latency for activity corresponding to faces without eyes, but PL activity still preceded ML (*Figure 2*; *Figure 2—figure supplement 1*), indicating that in the absence of eyes, face-specific activity still emerges across the ventral visual hierarchy in a bottom-up fashion.

To explore contextual associations of natural face experience, we then presented complex scenes that lacked faces, but contained cues that indicated where a face ought to be. When presented with images in which the face was occluded by some object in the image, surprisingly, the face cells in ML in both monkeys responded to the part of the image where the face ought to be, more than to non-face parts of the images (*Figure 3*). Though noisier than the population average, responses to occluded faces were apparent in individual channels (*Figure 3—figure supplement 1*). The magnitude and timing of responses to occluded faces varied across stimuli. Across all stimuli, the

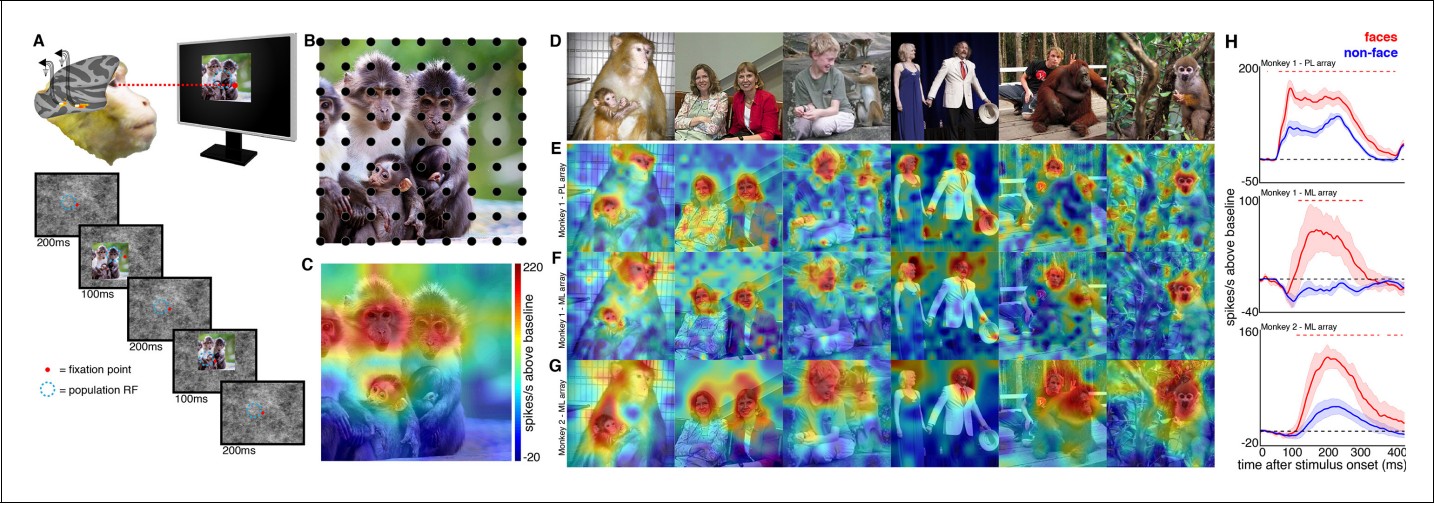

**Figure 1.** Experimental schematic and spatial selectivity of responsiveness. (**A**) The monkey maintains fixation (red spot) while a large (16˚ x 16˚) complex natural scene is presented at locations randomly sampled from a 17 × 17 (Monkey 1) or 9 × 9 (Monkey 2) grid of positions centered on the array's receptive field (blue dotted circle). On a given trial, 3 different images are presented sequentially at 3 different positions. Across the experiment, each image is presented multiple times at each position on the grid. (**B**) Parts of a 16˚ x 16˚ image (black dots) that were centered within each array's receptive fields across trials from the 9 × 9 stimulus presentation grid; position spacing, 2˚. (**C**) Map of a face cell's firing rate (100–250 ms after stimulus onset) in response to different parts of the image positioned over the receptive field (recording site 15 in Monkey 1 ML array). The most effective parts of this image were the monkey faces. (**D**) Complex natural images shown to the monkey. Population-level response maps to these images from (**E**) PL in Monkey 1; (**F**) ML in Monkey 1; and (**G**) ML in Monkey 2. Firing rates were averaged over 100–250 ms post stimulus onset for Monkey 1 and 150–300 ms for Monkey 2. For E, F, and G, the color scale represents the range between 0.5% and 99.5% percentiles of the response values to each image. All 3 arrays showed spatially specific activation to the faces in the images, with higher selectivity in the middle face patches. (**H**) PSTHs of responses to face (red) and non-face (blue) parts of these images for (top) Monkey 1 PL, (middle) Monkey 1 ML, and (bottom) M2 ML. Shading represents 95% confidence limits. Dashed red lines denote time window with a significant difference in response magnitude between face and non-face image parts (paired t-test across 8 images; t(7) > 2.64; p<0.05, FDR-corrected).

The online version of this article includes the following figure supplement(s) for figure 1:

**Figure supplement 1.** Localization of arrays.
**Figure supplement 2.** Receptive-field maps for all visually responsive sites in the two arrays in Monkey 1 (both in the right hemisphere) and the single array in Monkey 2 (left hemisphere).
**Figure supplement 3.** Category selectivity of all visually responsive sites in PL in one monkey (top) and ML in two monkeys (middle and bottom).
**Figure supplement 4.** Comparison of single-site response patterns with population average response patterns for all 3 arrays.
**Figure supplement 5.** Comparison of response patterns between simultaneously recorded PL and ML arrays in Monkey 1 for different time windows.

population-level activity to occluded faces was reliably higher than the average response to the rest of each image (t-test across 12 and 10 images for Monkeys 1 and 2, respectively; p<0.05, FDR-corrected). Interestingly, the responses to occluded faces on average were slower than the responses to intact faces by ~30–40 ms (*Figure 3—figure supplements 2* and *3*). The latency of responses to occluded faces was similar to that to eyeless faces (*Figure 2*; *Figure 2—figure supplement 1*). By 119 ms and 139 ms post stimulus onset for Monkey 1 and Monkey 2, respectively, the responses to occluded faces were significantly larger than the responses to the rest of the image. In contrast, responses to non-occluded versions of these faces were significantly larger than the responses to the rest of the image starting at 92 ms and 97 ms post stimulus onset for Monkey 1 and Monkey 2, respectively. It is possible that the face cells in ML responded to some tiny part of the head or neck that was still visible in each of the occluded-face photos, though similar visual features were present

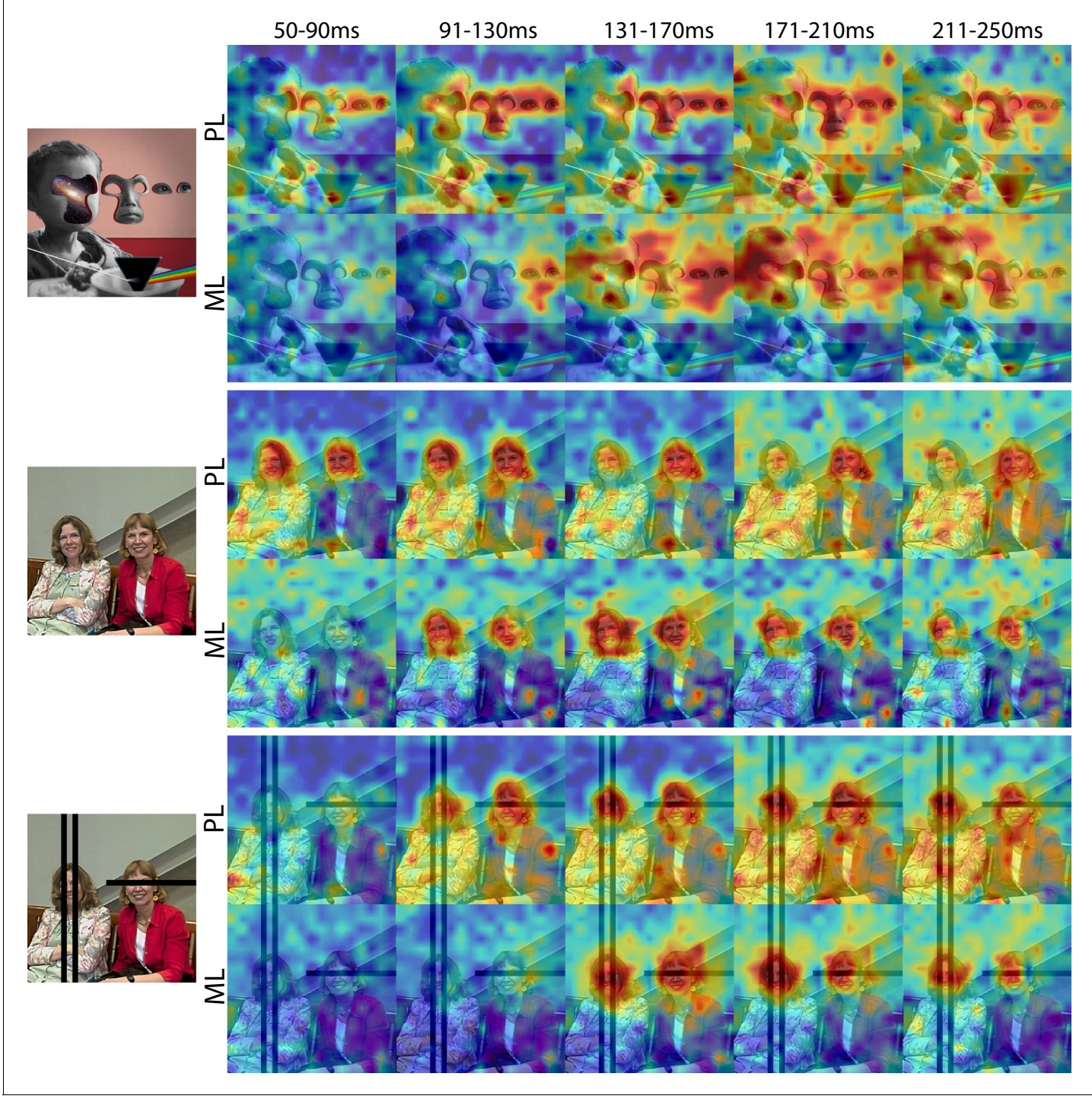

**Figure 2.** Cells in PL and ML responded to faces with and without eyes. Population-level response maps from Monkey 1 PL and ML for 40 ms bins between 50 and 250 ms post stimulus onset. Face-specific activity arises in PL prior to ML for images both with and without eyes, even though the latency for each depends on the presence of eyes. (top) Activity specific to eyes emerges earlier than responses to the rest of the face in both PL and ML. (middle and bottom) For both PL and ML, face activity emerges at longer latencies when the eyes are blacked out. Photo courtesy of Ralph Seigel. Response maps scaled to 0.5% and 99.5% percentiles of the response values to each image.

The online version of this article includes the following figure supplement(s) for figure 2:

**Figure supplement 1.** Comparison of response patterns to images of eyeless faces between PL and ML in Monkey 1.

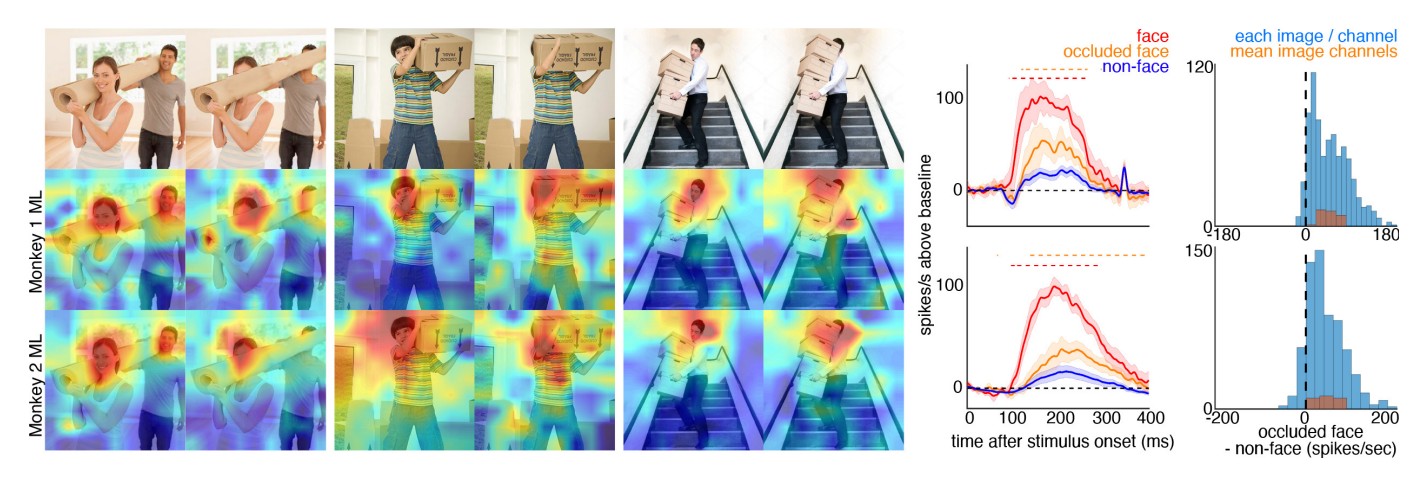

**Figure 3.** Cells in ML in both monkeys responded to occluded faces. (top) Images of occluded faces. Population-level response maps from (middle) Monkey 1 ML and (bottom) Monkey 2 ML. Firing rates were averaged over 100–250 ms post stimulus onset for Monkey 1 and 150–300 ms for Monkey 2. Response patterns scaled to 0.5% and 99.5% percentiles of the response values to each image. (right) PSTHs of responses in PL in Monkey 1 to regions of the image with occluded (orange) and non-occluded (red) faces and non-face parts (blue). Shading represents 95% confidence limits. (top) Responses of Monkey 1 ML. (bottom) Responses of Monkey 2 ML. Dashed red (and orange) lines denote time windows with significant differences in response magnitudes between face and occluded-face (and occluded vs. non-face) image parts (paired t-test across 12 images for Monkey 1; t(11) > 2.80 for occluded-face vs. non-face; t(11) > 2.70 for face vs. occluded-face; and 10 images for Monkey 2 t(9) > 2.56 for occluded-face vs. non-face; t(9) > 2.73; p<0.05, FDR-corrected). Note, the small spike just after 300 ms in Monkey 1 is a juicer artifact from the reward delivered after each trial and at least 200 ms prior to the onset of the next stimulus presentation. Histograms show response differences to occluded-face regions minus non-face control regions, for each image, each channel (blue) and for the mean image across channels (orange).

The online version of this article includes the following figure supplement(s) for figure 3:

**Figure supplement 1.** Individual site responses to occluded faces in ML for Monkeys 1 and 2.

**Figure supplement 2.** Responses to occluded faces in Monkey 1 ML for different time windows.

**Figure supplement 3.** Responses to occluded faces in Monkey 2 ML for different time windows.

in other parts of each image that did not evoke strong firing. It is also unlikely that the occluding objects contained visual features that alone evoke strong firing as those objects were present in other parts of each image that did not evoke strong firing (e.g., the section of the rug in-between the two faces and boxes covering the face of the man walking down the stairs vs. boxes below the face). Furthermore, when presented with people wearing masks, baskets or paper bags covering their faces, cells in ML responded to the masked faces (*Figure 4*). Similar to the occluded faces, responses to masked faces were also apparent in individual channels (*Figure 4—figure supplement 1*). The population-level responses to masked faces varied in magnitude and timing but were consistently stronger than average responses to the rest of each image (t-test across 10 and 11 images for Monkeys 1 and 2, respectively; p<0.05, FDR-corrected). The latency of responses to masked faces (*Figure 4—figure supplements 2* and *3*) was also similar to that to eyeless faces (*Figure 2*; *Figure 2—figure supplement 1*). Face cells in PL also responded more to occluded faces than to the non-face parts of the images (*Figure 4—figure supplement 4*; p<0.05, FDR corrected). Though cells in PL also responded on average more strongly to masked faces than to other parts of each image, this difference was not significant (*Figure 4—figure supplement 4*; p>0.05, FDR corrected). Thus, responses to occluded faces were more prominent at later stages of the ventral visual hierarchy.

To ask whether the face-cell responses to masked or occluded faces were due specifically to the context of a body just below the occluded face, we manipulated some images so that a non-face object was present in an image twice, once above a body, and once not (*Figure 5*, top). In both monkeys' ML patches, responses to the non-face object positioned above a body were larger than responses to the same object when not above a body, both at the population level (*Figure 5*) and in individual channels (*Figure 5—figure supplement 1*). Similar to the occluded and masked faces, population-level responses to these face-swapped images varied in magnitude and latency, but responses to non-face objects were consistently stronger when positioned above a body than when

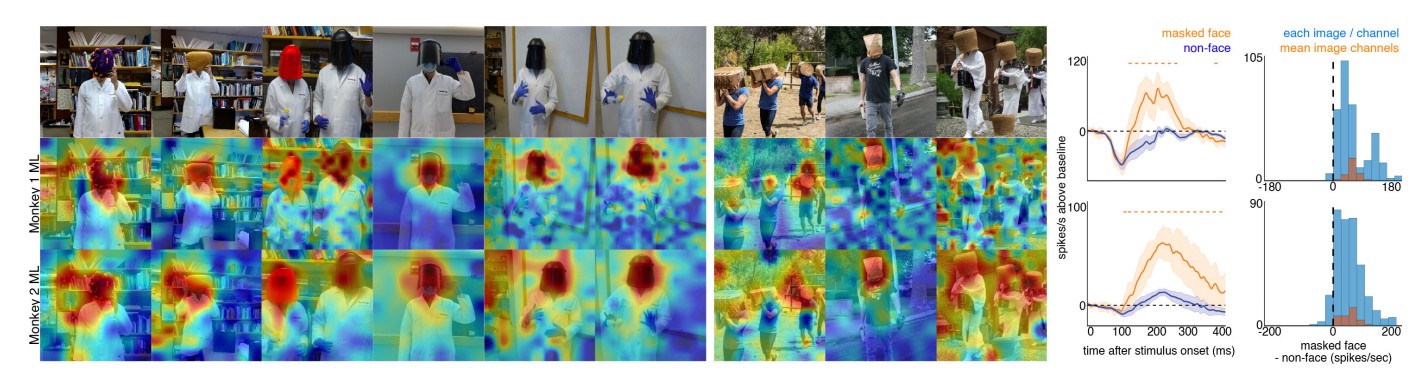

**Figure 4.** Responses to masked faces. Cells in ML in both monkeys responded to faces covered by masks, baskets, or paper bags. (top) Images of covered faces. Population-level response maps from (middle) Monkey 1 ML and (bottom) Monkey 2 ML. Firing rates were averaged 100–250 ms post stimulus onset for Monkey 1 and 150–300 ms for Monkey 2. Response patterns scaled to 0.5% and 99.5% percentiles of the response values to each image. (right) PSTHs of responses of sites in ML in Monkeys 1 and 2 to covered faces (orange). Shading represents 95% confidence limits. (top) Responses of sites in Monkey 1 patch ML. (bottom) Responses of sites in Monkey 2 ML. Dashed orange lines denote time windows with significant differences in response magnitudes between covered-face vs. non-face image parts (paired t-test across 10 images for Monkey 1 and 11 images for Monkey 2; t(9) > 2.81 for Monkey 1; t(10) > 2.42 for Monkey 2; p<0.05, FDR-corrected). Histograms show response differences to masked face minus non-face control regions, for each image, each channel (blue) and for the mean image across channels (orange).

The online version of this article includes the following figure supplement(s) for figure 4:

**Figure supplement 1.** Individual channel responses to masked faces in Monkey 1 and 2 ML.
**Figure supplement 2.** Responses to masked faces in Monkey 1 ML for different time windows.
**Figure supplement 3.** Responses to masked faces in Monkey 2 ML at different time windows.
**Figure supplement 4.** Responses to faces and masked faces in Monkey1 PL.

positioned elsewhere (t-test across 8 and 15 images for M1 and M2, respectively; p<0.05, FDR-corrected). Similar to the occluded and masked faces, the latency of responses to the non-face object above a body (*Figure 5—figure supplements 2* and *3*) was comparable to that to eyeless faces (*Figure 2*; *Figure 2—figure supplement 1*). Though cells in PL also responded on average more strongly to objects on top of bodies, this difference was not significant (*Figure 4—figure supplement 4*; p>0.05, FDR-corrected). Thus, responses in ML to a non-face image were facilitated by a body below it.

Given our finding that bodies facilitated face-cell responses to masked, occluded, and non-face objects positioned above them, we might expect bodies to also facilitate responses to faces appropriately positioned above bodies, but this was not the case. Instead, responses to faces above bodies were indistinguishable from responses to disembodied faces (*Figure 6*, leftmost; *Figure 6—figure supplements 1* and *2*, top two rows). This is consistent with the response maps to a subset of images that showed strong responses to faces regardless of the spatial relation to bodies (columns 3–5 in *Figure 5*, top). In contrast, responses to face-shaped noise patches (i.e. where face pixels were replaced by random 3-color channel pixel values), face outlines, and non-face objects were facilitated by the presence of a body positioned below (*Figure 6*; *Figure 6—figure supplements 1* and *2*). Face-shaped noise patches and face-shaped outlines both elicited responses above baseline even in the absence of the body (middle two columns). This was likely due to these images containing typical shape features of a face, which is sufficient to drive these neurons (*Issa and DiCarlo, 2012*; *Tsao et al., 2006*). Importantly, the responses to these images when placed above a body were significantly larger (p<0.05, FDR-corrected). Further, non-face objects activated these neurons only when placed above bodies (*Figure 6* rightmost column, p<0.05, FDR-corrected; *Figure 6—figure supplements 1* and *2*, bottom row). Together, these data suggest a ceiling effect–that bodies facilitate responses of face cells only when the presence of a face is ambiguous. Further, these neurons respond to the region above a body even if no foreground object is present (*Figure 6—figure supplement 3*). In the absence of a face, the presence of a body below the receptive field appears to be sufficient for driving these face-selective cells.

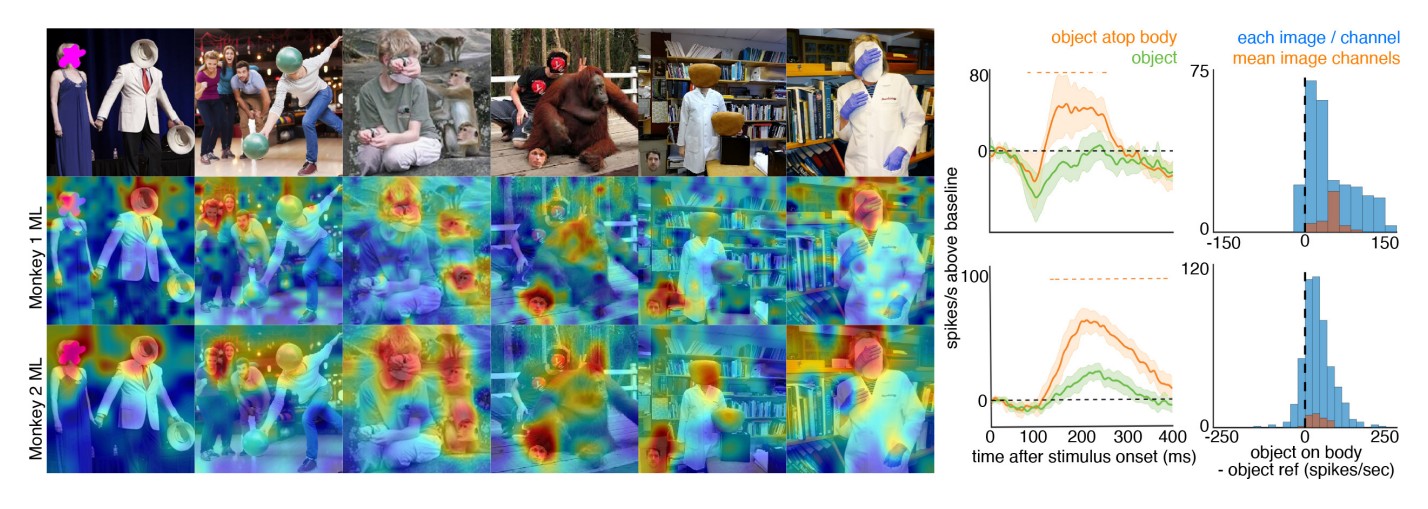

**Figure 5.** Face-like responses to non-face objects. (top) Manipulated images with non-face objects in positions where a face ought and ought not to be. Population-level response maps from (middle) Monkey 1 ML and (bottom) Monkey 2 ML. Firing rates were averaged 100–250 ms post stimulus onset for Monkey 1 and 150–300 ms for Monkey 2. Response patterns scaled to 0.5% and 99.5% percentiles of the response values to each image. (right) PSTHs of Monkey 1 PL to non-face images above a body (orange) or not above a body (green). (top) PSTHs from Monkey 1 ML. (bottom) PSTHs from Monkey 2 ML. Shading indicates 95% confidence intervals. Dashed orange lines denote time windows with significant differences in response magnitudes between object-atop-body vs. object-apart-from-body (paired t-test across 8 and 15 images for Monkey 1 and 2, respectively; t(7) > 3.04 for Monkey 1; t(14) > 2.38 for Monkey 2; p<0.05, FDR-corrected). Histograms show response differences to object atop body minus object control regions for each image, each channel (blue) and for the mean image across channels (orange).

The online version of this article includes the following figure supplement(s) for figure 5:

**Figure supplement 1.** Individual channel face-like responses to non-face objects in Monkey 1 and 2 ML.
**Figure supplement 2.** Face-like responses to non-face objects in Monkey 1 ML for different time windows.
**Figure supplement 3.** Face-like responses to non-face objects in Monkey 1 ML for different time windows.

If bodies positioned directly below the activating region facilitate face-cell responses, is it because face cells have a discrete excitatory region, below the activating region, that is simply responsive to bodies? Or is the effect more complex, reflecting contextual information that a body should be accompanied by a face? To address this question, we mapped responsiveness to bodies, with and without heads, at varying orientations and locations in a 9 × 9 grid pattern (*Figure 7—figure supplement 1*). For bodies-with-heads, responses were maximal when the face was centered in the cells' activating region, for each orientation (*Figure 7A*, top row). Headless bodies produced activation maps that were similarly maximal at positions in which a face ought to be centered on the cells' activating region, not just when the body was positioned below the activating region (*Figure 7A*, bottom two rows). The magnitude of firing rate varied across the tested visual field locations (e.g., firing rates tended to be higher in the upper half of the image). Composite maps show the preferred body orientation for each spatial position tested (*Figure 7A*, right side). In experiments both with and without heads, preferred orientation systematically varied such that the preferred body orientation at that spatial location positioned the face, or where the face ought to be, within the RF center. Interestingly, these neurons were also responsive when the feet landed in the activating region, for the bodies without heads, and, to a lesser extent, the bodies with heads. This was especially apparent for inverted bodies, suggesting that both end-of-a-body and above-a-body are cues that a face might be expected. The surprising result that face cells are maximally responsive to where a face would be expected, largely irrespective of the body orientation, indicates that the responses to occluded faces documented in *Figures 3* and *4* are not responses merely to bodies located below the cells' face-selective regions, but instead reflect information about where a face ought to be, for a given body configuration.

If bodies convey information to face cells about where a face ought to be, are there preferred face-body configurations? When face and face-shaped noise were presented attached to bodies in upright and inverted configurations, the strongest response was when the face (or face-shaped

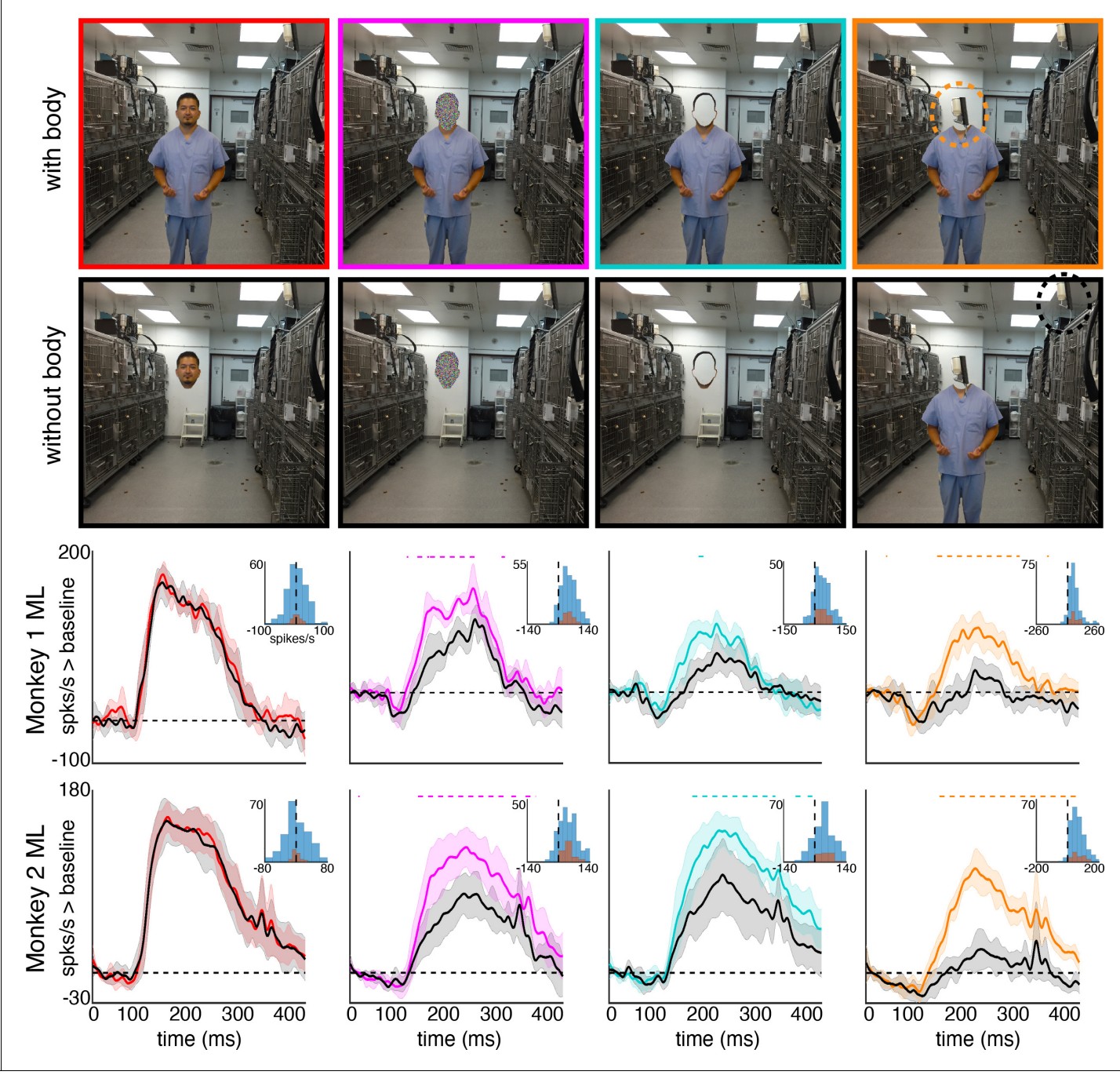

**Figure 6.** Body-below facilitation of face-cell responses to non-face objects but not to faces. Images of face, noise, face outline, and non-face object (top row) above a body and (second row) without a body. (bottom two rows) PSTHs from ML in Monkeys 1 and 2. Responses were calculated over the same face-shaped region for all images, for 7 different image sets of different individuals. Shading indicates 95% confidence intervals. Dashed colored lines denote time windows with significant differences in response magnitudes between face region above body vs. without body (paired t-test across 7 images; t(6) > 2.79 for noise; t(6) > 3.08 for outlines; t(6) > 2.76 for non-face objects; p<0.05, FDR-corrected). Inset histograms show response differences to the regions atop body minus the corresponding regions of no-body images, for each image, each channel (blue) and for the mean image across channels (orange).

The online version of this article includes the following figure supplement(s) for figure 6:

**Figure supplement 1.** Responses to face and non-face images with and without bodies.
**Figure supplement 2.** Responses to face and non-face images with and without bodies.
**Figure supplement 3.** Responses to the region above bodies in the absence of faces or heads.

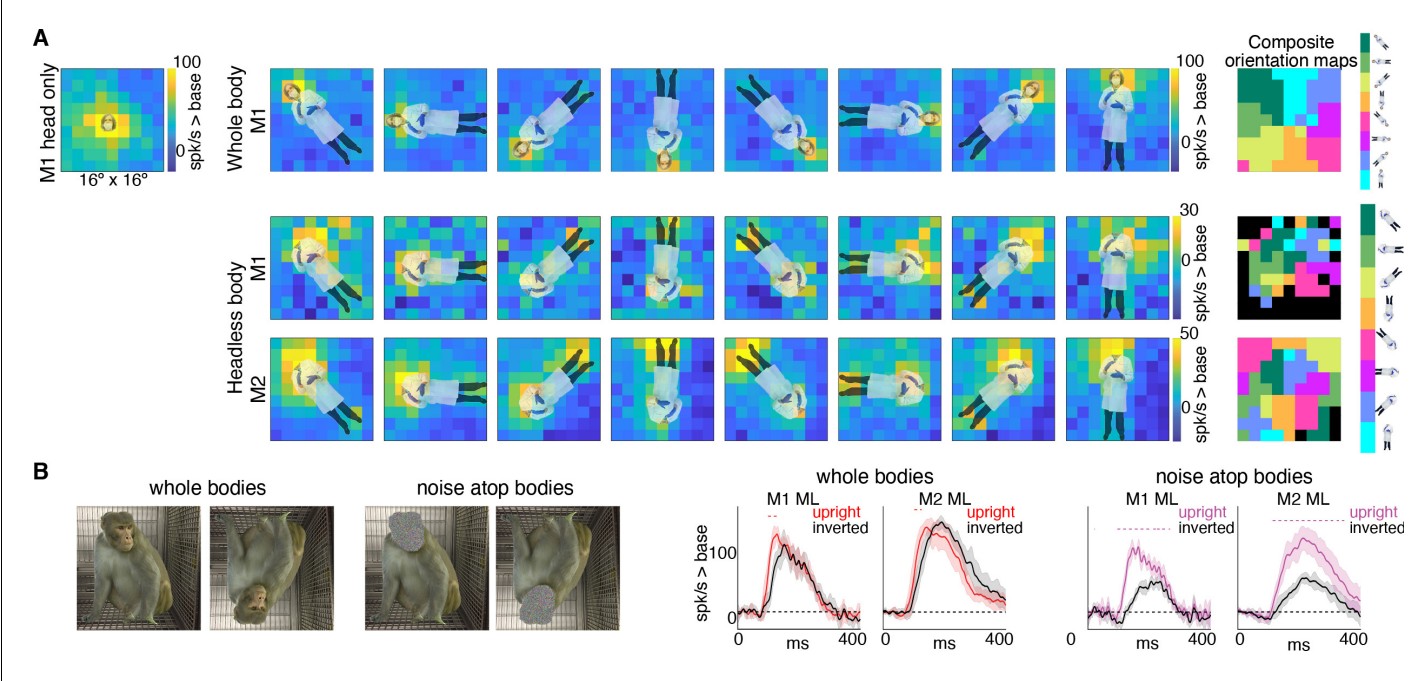

**Figure 7.** Body-orientation selectivity. (**A**) Upper left panel shows an example head-only image overlaid on the population-level activation map averaged across all head only images (n = 8) from Monkey 1 ML. The upper row shows an example whole body image at eight different orientations each overlaid on the average activation maps for all whole bodies (n = 8); maximum responsiveness was focal to the face at each orientation. The lower two rows show a similar spatial selectivity for the region where a face ought to be at each orientation for headless bodies (n = 8) in Monkey 1 and 2 ML. Whole and headless body image sets were presented on a uniform white background (*Figure 7—figure supplement 1*). The composite maps on the far right show the best body orientation at each grid point for whole bodies and headless bodies for face patch ML in both monkeys, as indicated. The scale of the body orientations from top to bottom corresponds to the body orientations shown left to right in the response maps. Black sections denote areas with responses at or below baseline. For both whole and headless bodies, maximal activation occurred for bodies positioned such that the face (or where the face ought to be) would be centered on the cells' activating region. (**B**) PSTHs when whole body and face-shaped noise atop bodies were presented within the cell's activating region in both monkeys. For whole body images, there was only a brief difference in activity between upright and inverted configurations during the initial response. For face-shaped noise, the response was substantially larger for the upright (vs. inverted) configurations for the duration of the response. Dashed colored lines denote time windows with significant differences in response magnitudes between upright vs. inverted face regions of the image (t-test across 7 images; t(6) > 4.74 for intact faces; t(6) > 3.34 for noise; p<0.05, FDR-corrected). The online version of this article includes the following figure supplement(s) for figure 7:

**Figure supplement 1.** Body-orientation experiment.

**Figure supplement 2.** Responses to upright and inverted face and noise images.

**Figure supplement 3.** Stimuli used in each experiment and the corresponding binary ROIs of the image regions used for calculating responses or psths corresponding to faces or regions above the body.

noise) was presented at the center of the activating region (*Figure 7B*). For intact faces with bodies presented at the center of the activating region, there was only a small inversion effect (*Figure 7B*; whole bodies). While reaction time differences between upright and inverted faces are robust and reliable across studies (*Leehey et al., 1978*; *Yin, 1969*), differences in response magnitudes within IT face domains are modest or not consistently found in humans (*Aguirre et al., 1999*; *Haxby et al., 1999*; *Kanwisher et al., 1998*) or monkeys (*Bruce, 1982*; *Pinsk et al., 2009*; *Rosenfeld and Van Hoesen, 1979*; *Taubert et al., 2015*; *Tsao et al., 2006*). The lack of a strong inversion effect for intact-faces-over-bodies compared to a stronger inversion effect for non-face-objects-above-bodies could reflect a ceiling effect, or it could mean that, in the presence of a face, tuning is more flexible and more invariant to spatial transformations. In contrast to responses to intact faces, there was a large, sustained difference in response magnitude between upright and inverted face-shaped noise patches with bodies (*Figure 7B*; noise atop bodies), suggesting that the body facilitation is maximal when the presence of a face is ambiguous and above a body. The preference for upright vs. inverted configurations with the face-shaped noise is consistent with biases apparent in *Figure 7A*. Though

face cells in ML were responsive to headless bodies at a variety of spatial positions, activity tended to be strongest when bodies were positioned below the activating region (*Figure 7A*; bottom two rows). This was particularly clear in Monkey 2 where the part of the image that gave the strongest response for the inverted (180-degree) body orientation was the feet, meaning that the inverted body still drove neurons most when it was positioned just below the activating region. Thus, for face-ambiguous stimulation, contextual responses are strongest to spatial configurations that reflect typical experience.

How can face-selective neurons respond to bodies positioned below their receptive fields? One possibility is that such contextual sensitivity is intrinsic to IT and pre-specified for the processing of biologically important information. Alternatively, given that the development of face domains depends on seeing faces during development (*Arcaro et al., 2017*), these conjoint representations might also be learned over development as a consequence of frequently seeing faces and bodies together. To explore whether learning is sufficient to link face and body representations, we analyzed the representation of image categories in a computational model that has similar architectural features to the object processing pathway in primates. Specifically, we took a convolutional neural network, AlexNet (*Krizevsky et al., 2012*), trained to classify natural images into 1000 object categories, and assessed its activation patterns evoked by images of isolated objects from 6 different categories; human faces, headless human bodies, animals, houses, inanimate objects, and handheld tools (*Figure 8A*, top). We tested a stimulus set widely used for localizing category-selective regions in IT cortex (*Downing et al., 2006*). We assessed the similarity of activation patterns between all image pairs in every layer of AlexNet and in pixel space. As seen in the multidimensional scaling visualization, images are distinguished along categorical distinctions even based on just pixel intensity (*Figure 8A*, bottom). The Euclidean distances between image pairs within each category were smaller than between categories, indicating that images were more homogenous within than between categories and highlighting that low-level features such as shape, color, or texture are a major component of these commonly tested 'semantic' categories (also see *Dailey and Cottrell, 1999*). Distinctions between categories were present in early layers of AlexNet (*Figure 8A*, bottom). The distinction between faces and non-faces persisted across all layers of AlexNet. The mean distance between face pairs (red) was smaller than the mean distance between faces and non-body,

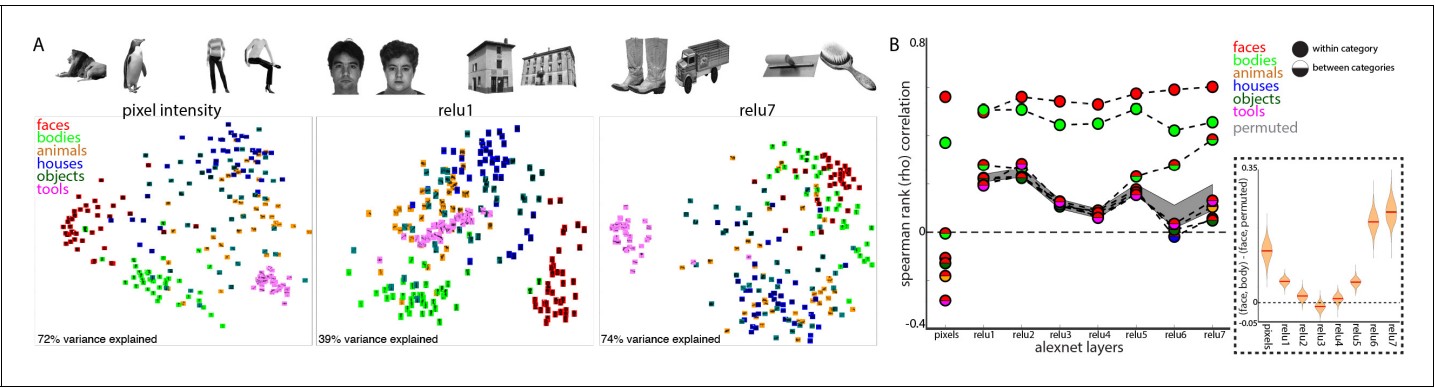

**Figure 8.** Learned face and body representations in hierarchical artificial neural network (AlexNet). Representational similarity of a stimulus set comprising 240 images across 6 categories including faces and bodies. (**A**, top) Two example images from each category. (**A**, bottom) Visualization of similarity between faces (red), bodies (green), houses (blue), objects (green), and tools (pink) from multidimensional scaling of (left column) pixel intensity, (middle column) relu1 units, (right column) relu7 units) Euclidean distances. (**B**) Comparison of representational similarity within and between categories. Spearman rank correlations were calculated between image pairs based on pixel intensity and activations within AlexNet layers for all object categories. Mean similarity within (solid filled circles) and between (dual colored circles) categories are plotted for pixel intensity and the relu layers of AlexNet. The 2.5% and 97.5% percentiles from a permutation test in which the non-face stimulus labels were shuffled before calculating correlations (grey shaded region) are plotted for pixel and relu layers of AlexNet comparisons. Across all layers, both face and body images are more similar within category than between categories. (inset) The representations of faces and bodies are most similar to each other in deep layers (relu6 and relu7) of AlexNet.

The online version of this article includes the following figure supplement(s) for figure 8:

**Figure supplement 1.** Comparison of correlations between trained and untrained versions of AlexNet.

non-face categories (animals, houses, objects, and tools). Across all rectified linear unit (relu) layers of AlexNet, activation patterns were more similar for within category comparisons of both faces and bodies (*Figure 8B*, solid red and green circles, Spearman rank correlations) than between faces and other categories (dual colored circles). At the pixel level and in early layers of AlexNet, the mean similarity between faces and bodies (half red/half green circles) was greater than the similarity between faces and the other categories; this similarity was also stronger to that obtained under randomization of the stimulus labels for non-face categories (this randomization distribution is marked by the grey shaded region that denotes 2.5% and 97.5% bounds). In the deeper layers of AlexNet (relu6 and relu7), face and body image representations become more overlapping (*Figure 8B*, inset). In relu7, the correlation between face and body image pairs were only slightly weaker than the correlation between pairs of body images. Comparable effects were observed when assessing similarity with Pearson correlations and Euclidean distance on z-score normalized data. Further, these results were specific to the trained AlexNet, as untrained versions of the network did not yield strong face-body correlations and these correlations were no different than correlations between faces and other categories (*Figure 8—figure supplement 1*). These results demonstrate that representations of faces and bodies can merge over learning without any explicit training about faces and bodies. This is particularly impressive given that AlexNet was not trained to categorize faces, bodies, or even humans. Because faces and bodies co-occur in people and people were present in the training images, faces and bodies share variance in their informativeness with respect to class labels. There is thus pressure for the network to use overlapping representations in the face and body hidden layer representations in this supervised learning scenario. This phenomenon should hold more generally in unsupervised or self-supervised settings as the presence of image co-occurrence in the input would be expected to lead to similar overlap. Though convolutional networks can capture some aspects of the representational structure of biological networks (*Hong et al., 2016*; *Ponce et al., 2019*), these networks as currently implemented are not well suited for probing the retinotopic specificity of object representations, because they are explicitly constrained to process inputs from different portions of an image in the same way. Future work could probe the spatial relationships between commonly experienced object categories in computational models more optimized for maintaining spatial coding.

## Discussion

Our finding that face-selective neurons respond to parts of an image where a face ought to be—above a body—goes well beyond previous observations on category selectivity in inferotemporal cortex. Previous neuronal recordings in macaques found that face-selective neurons respond only weakly to images of headless bodies, and body-selective neurons poorly to faces, suggesting that face and body circuits are largely independent (*Freiwald and Tsao, 2010*; *Freiwald et al., 2009*; *Kiani et al., 2007*; *Tsao et al., 2006*). Human fMRI studies have also provided evidence that face and body processing are independent (*Kaiser et al., 2014*; *Peelen and Downing, 2005*; *Peelen et al., 2006*; *Schwarzlose et al., 2005*) and that face-selective domains in the FFA do not respond to headless animals or human bodies presented at fixation (*Kanwisher et al., 1999*; *Peelen and Downing, 2005*; *Schwarzlose et al., 2005*). Consistent with these prior findings, the face neurons we recorded from responded strongly and selectively to faces presented over a limited part of the visual field but did not respond to bodies presented to the same region. However, these neurons also responded to masked and occluded faces that lacked intrinsic facial features, but were above bodies. Further, when non-face objects that did not by themselves evoke a response from these neurons were placed above a body, the face-selective neurons fired robustly. Consistent across all six experiments, these neurons responded to bodies positioned such that the body implied the presence of a face within the neurons' activating regions. In prior studies, both headless bodies and faces were presented at fixation, so our results suggest that these studies might have found larger responses to bodies if the bodies had been presented lower in the visual field, below the center of maximal activation to faces. Thus, face cells demonstrate a large contextual surround tuned to entities that normally co-occur with faces.

Though prior studies have mostly provided evidence for the independence of face- and body-processing circuits, a few fMRI studies found facilitatory influences of bodies on blood-flow responses in face-selective brain regions: The face fusiform area (FFA) in humans has been reported

to be responsive to animal bodies with obscured faces (*Chao et al., 1998*) or to blurred faces above bodies (*Cox et al., 2004*). *Cox et al., 2004* also found body-below facilitation of BOLD signal for blurred faces but not for clear faces, and they found that the BOLD signal facilitation depended on the face and body being in a correct spatial configuration. Both these findings are consistent with our neuronal recording results. In monkeys, the most anterior face patches show stronger fMRI activations to images of whole monkeys than to the sum of the responses to faces and to bodies; however, the middle and posterior face patches (which we recorded here) do not show facilitation, and have only sub-additive responses (*Fisher and Freiwald, 2015*). Thus, although there is some fMRI evidence for interactions between face and body processing, fMRI studies are limited in that they can measure responses only of large populations of neurons of potentially mixed selectivity and therefore cannot resolve the relationship of face and body neural circuits. Here we show by direct neuronal recording that face and body circuits are not independent and that face-selective neurons can express large, complex, silent, contextual surround effects. From simultaneously recording neurons in PL and ML face patches, we show that these responses emerge fast and are first detected in the posterior patch, suggesting that contextual effects involve at least some feedforward inputs that cannot be attributed to cognitive processes such as mental imagery or expectation. The idea that the context of a visual scene leads to the expectation, enhanced probability, or prior of seeing some particular item is usually thought to involve top-down processes. However, if the feedforward inputs to, say, face-selective neurons, are sculpted by experience to be responsive not only to faces, but also to things frequently experienced in conjunction with faces, then such expectations would be manifest as the kind of complex response properties reported here. Furthermore, in contrast to the traditional view of a dichotomy of spatial and object vision, our data demonstrate that object processing in IT is spatially specific (also see *Op De Beeck and Vogels, 2000*).

More broadly, these results highlight that neurons in inferotemporal cortex do not code objects and complex shapes in isolation. We propose that these neurons are sensitive to the statistical regularities of their cumulative experience and that the facilitation of face-selective neurons in the absence of a clear face reflects a lifetime of experience that bodies are usually accompanied by faces. Recently, we proposed that an intrinsic retinotopic organization (*Arcaro and Livingstone, 2017*; also see *Hasson et al., 2002*) guides the subsequent development of category-selectivity in inferotemporal cortex, such as face and body domains (*Arcaro et al., 2017*). For example, monkeys normally fixate faces, and they develop face domains within regions of inferotemporal cortex that represent central visual space (*Figure 9*, top). Here, we propose that these same developmental mechanisms can account for contextual learning. Our results indicate that, just as the retinotopic regularity of experiencing faces guides where in IT face domains develop, the spatial regularity between faces and bodies is also learned in a retinotopic fashion and may account for why face and body patches stereotypically emerge in adjacent parts of cortex (*Figure 9*, bottom) in both humans (*Schwarzlose et al., 2005*) and monkeys (*Bell et al., 2009*; *Pinsk et al., 2009*). Such a spatial regularity is consistent with prior observations of a lower visual field bias for the RF centers of body-selective neurons near ML (*Popivanov et al., 2015*) and an upper visual field bias for eye-selective neurons in PL (*Issa and DiCarlo, 2012*). Thus, something as complex and abstract as context may be firmly grounded in the ubiquitous mechanisms that govern the wiring-up of the brain across development (*Arcaro et al., 2019*).

## Materials and methods

All procedures were approved in protocol (1146) by the Harvard Medical School Institutional Animal Care and Use Committee (IACUC), following the *Guide for the care and use of laboratory animals* (Ed 8). This paper conforms to the ARRIVE Guidelines checklist.

### Animals and behavior

Two adult male macaques (10–12 kg) implanted with chronic microelectrode arrays were used in this experiment. In monkey 1, two floating microelectrode arrays were implanted in the right hemisphere superior temporal sulcus – one within the posterior lateral (PL) face patch in posterior inferotemporal cortex (PIT) (32-channel Microprobes FMA; https://microprobes.com/products/multichannel-arrays/fma) and a second array within the middle lateral (ML) face patch in central inferotemporal cortex (CIT) (128-channel NeuroNexus Matrix array; http://neuronexus.com/products/primate-large-brain/

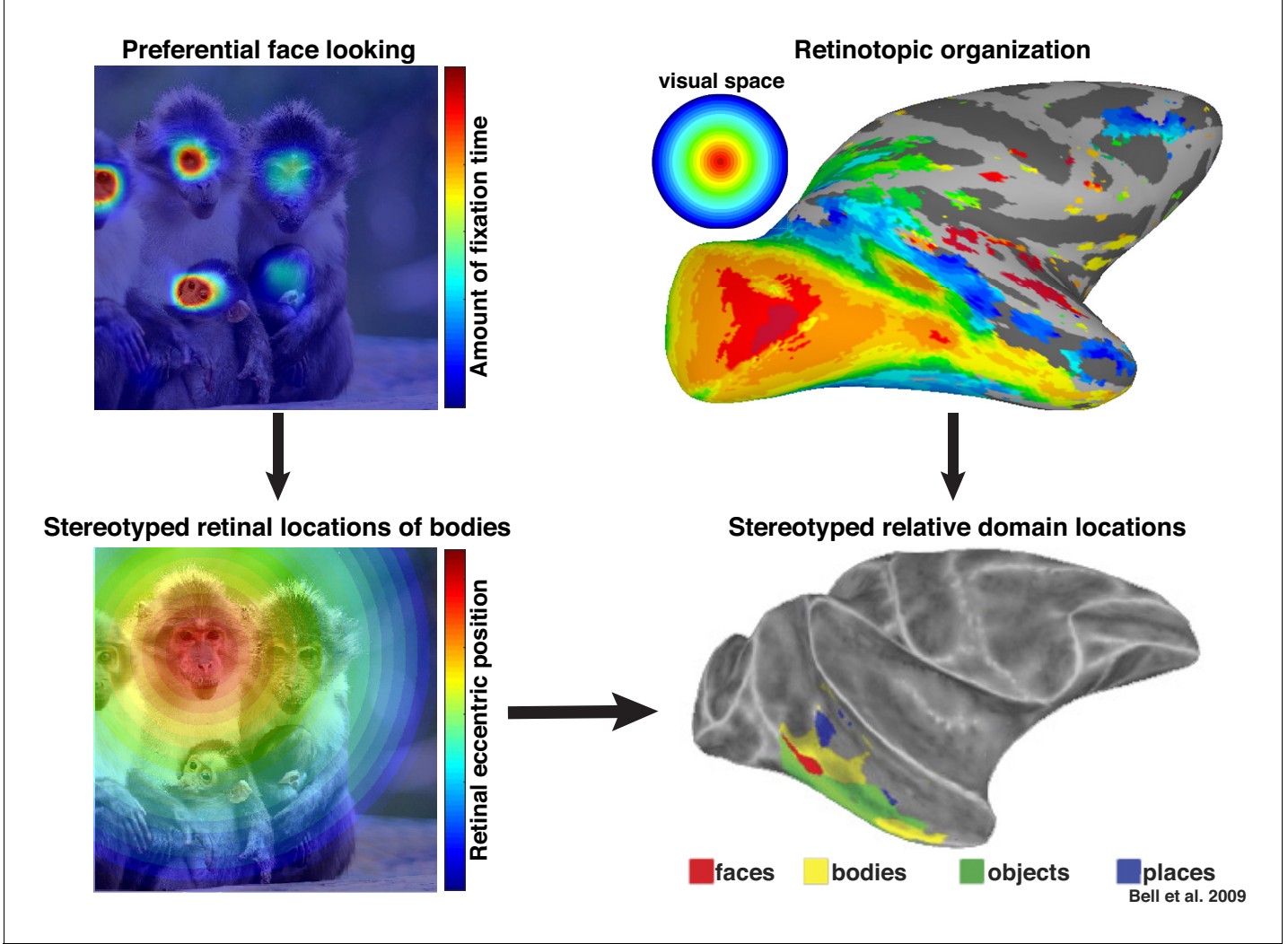

**Figure 9.** Preferential looking at faces imposes stereotyped retinal experience of bodies. (top left) Primates spend a disproportionate amount of time looking at faces in natural scenes. (top right) This retinal input is relayed across the visual system in a retinotopic fashion. (bottom left) Example retinal experience of a scene along the eccentricity dimension when fixating a face. Given that bodies are almost always experienced below faces, preferential face looking imposes a retinotopic regularity in the visual experience of bodies. (bottom right) This spatial regularity could explain why face and body domains are localized to adjacent regions of retinotopic IT.

matrix-array-primate/). In monkey 2, one 32-channel floating microelectrode array was implanted in left hemisphere ML. Monkeys were trained to perform a fixation task. Neural recordings were performed on a 64-channel Plexon Omniplex Acquisition System. For monkey 1, recordings were performed simultaneously for the 32-ch PIT array and 32-ch from the CIT array. In both monkeys, the same channels were recorded from for each experimental session. The internal dimensions of the Microprobes array was 3.5 × 1.5 mm with a spacing of 370–400 microns between electrodes (across the width and length of the array respectively). The Neuronexus matrix array was a comb arrangement, with 200 micron spacing between the 4 shanks and 400 microns site spacing along the shanks. The arrays were oriented with the long axis AP. The task required the monkey to keep their gaze within ± 0.75° of a 0.25° fixation spot. We used an ISCAN eye monitoring system to track eye movements (www.iscaninccom).

## fMRI-guided array targeting

Functional MRI studies were carried out in both monkeys to localize face patches along the lower bank of the superior temporal sulcus. For scanning, all monkeys were alert, and their heads were immobilized using a foam-padded helmet with a chinstrap that delivered juice. The monkeys were scanned in a primate chair that allowed them to move their bodies and limbs freely, but their heads were restrained in a forward-looking position by the padded helmet. The monkeys were rewarded with juice during scanning. Monkeys were scanned in a 3 T TimTrio scanner with an AC88 gradient insert using 4-channel surface coils (custom made by Azma Mareyam at the Martinos Imaging Center). Each scan session consisted of 10–12 functional scans. We used a repetition time (TR) of 2 s, echo time (TE) of 13 ms, flip angle of 72°, iPAT = 2, 1 mm isotropic voxels, matrix size 96 × 96 mm, 67 contiguous sagittal slices. To enhance contrast (*Leite et al., 2002*; *Vanduffel et al., 2001*), we injected 12 mg/kg monocrystalline iron oxide nanoparticles (Feraheme, AMAG Pharmaceuticals, Cambridge, MA) in the saphenous vein just before scanning. Responses to image categories of faces and inanimate objects were probed. Brain regions that responses more strongly to faces than objects were identified along the lower bank of the STS (p<0.0001, FDR-corrected). *Arcaro and Livingstone, 2017* for additional details.

After array implantation, Computed Tomography (CT) scans (0.5 × 0.5×1.25 mm) were collected in both monkeys. The base of the FMA arrays and wires were visible on both CT images. The base of the NeuroNexus array was not clearly visible on the CT image. Each monkey's CT image was spatially aligned to its MPRAGE anatomical image. Because brain/skull contrast is opposite between CT and MPRAGE MRI images, the two volumes were aligned by manually creating a binary brain mask for both CT and MPRAGE images and rigidly aligning the brain masks. The resulting spatial transformation matrix was applied to bring the CT and MPRAGE images into alignment. The location of the FMA arrays were then compared with the fMRI-defined PL and ML face patches.

## Electrophysiology display

We used MonkeyLogic to control experimental workflow (https://monkeylogic.nimh.nih.gov). Visual stimuli were displayed on an 19' Dell LCD screen monitor with a 60 Hz refresh rate and a 4:3 aspect ratio positioned 54 cm in front of each monkey.

## Receptive field mapping

To map the location of each recording site's receptive field, we flashed small (~2°x2°) images of faces or round non-face objects at different locations relative to the fixation point in a 16°x16° grid with 1–2° spacing centered at the fixation point. Each image was presented for 100 ms ON and 200 ms OFF. Responses were defined as the mean firing rate over 80–250 ms after picture onset minus the mean firing rate over the first 30 ms after image onset. Responses were averaged across image repetitions. A 2-dimensional gaussian was fit to each array channel's 9 × 9 response grid to derive the centroids and response widths (1 sigma). We define the population RF as the mean array response. To characterize tuning of each recording site, images of isolated faces, hands, bodies, and objects on a white background were presented within the activating region of all the visually-responsive sites. Each image subtended 4° and was presented for 100 ms ON and 200 ms OFF. Responses were defined as the mean firing rate over 80–250 ms after image onset minus the mean firing rate over the first 30 ms after image onset. Responses were averaged across image repetitions. Category selectivity was assessed with an index (category1-response –category2-response)/ (category1-response + category2-response) (*Baker et al., 1981*). For computing the index, channels with negative responses were excluded. There were no channels with negative responses in the PL array in Monkey 1. For the ML arrays in both monkeys, we excluded 1 channel each from the body vs objects index due below baseline responses to bodies.

## Main Experiment stimuli

Stimuli comprised 16° x 16° natural scene photographs embedded in a brown-noise image that filled the entire monitor screen. Each photo could contain humans, monkeys, other animals, or artificial objects (*Figure 7—figure supplement 3*). Several of the photos were modified to remove the face or to replace the face with non-face objects or pixelated noise. Photos were grouped into several image sets. The first image set comprised unaltered natural scenes containing humans and monkeys.

The second image set comprised images of people wearing masks. The third image set contained images where faces were occluded with objects. The fourth image set comprised images where objects in each scene were pasted over the faces. The fifth image set comprised 6 people and one monkey with 8 variations for a total of 56 images. For each person/monkey, the face was replaced with either pixelated noise (i.e. where face pixels were replaced by random 3-color channel pixel values), an outline, another object in the scene, or left intact. Each face manipulation was presented either above a body or without the body. The sixth image set comprised 7 people and 1 monkey each at one of 8 orientations at 45° intervals for a total of 64 images. The seventh image set comprised 6 people and 1 monkey with 4 variations for a total of 28 images. For each person/monkey, the face was presented either intact or replaced with pixelated noise and either in upright or inverted orientation. For each experiment, pictures were presented over a 17 x 17 point grid (spacing 1°) for Monkey 1 and a $9 \times 9$ point grid (spacing of 2°) for Monkey 2 with the image centered on the center of the population receptive field. Generally, Monkey 2 did not work for as long as Monkey 1 so a coarser grid was necessary in order to get enough repetitions of all stimuli at each position. The difference in grid sampling is unlikely to be a major factor as results for all experiments were consistent between monkeys. Further, the 2° spacing of this $9 \times 9$ grid remained smaller than the response width of the RF mapping and was sufficient for detecting spatially specific activity. Images were presented for 100 ms ON and 200 ms OFF. Three images were presented per trial (Figure 1A). At the start of each trial, the fixation target appeared, and the animal had up to 5 s to direct its gaze to the fixation target. If the animal held fixation until the end of the trial, a juice reward was delivered at 100 ms after the last (third) image's offset period. Every picture was presented at each grid position 5–10 times. In Monkey 1, PL and ML arrays were recorded from simultaneously.

## Experimental session workflow

Data were collected across 4 months in M1 and 1 month in M2. Each day, the animal would be head-fixed and its implant(s) connected to the experimental rig. First, the animal's gaze was calibrated with a 9-point grid using a built-in MonkeyLogic routine. We used the Plexon Multichannel Acquisition Processor Data Acquisition System to collect electrophysiological information, including high-frequency ('spike') events, local field potentials, and other experimental variables, such as reward rate and photodiode outputs tracking monitor frame display timing. Each channel was auto-configured daily for the optimal gain and threshold; we collected all electrical events that crossed a threshold of 2.5 SDs from the mean peak height of the distribution of electrical signal amplitudes per channel. These signals included typical single- and multi-unit waveforms.

## Spike data preparation

The raw data comprised event ('spike') times per channel for the entire experimental session. In Monkey 1, 32 channels were recorded simultaneously from both the posterior and middle face patch arrays for the main experiment. The same channels were recorded from for the entire duration of this study.

## Analysis

For each image, trials were sorted based on grid position. Trial repetitions were averaged to derive the mean activity when each grid position was centered within the mean activating region of each array. The mean activity was visualized over each image with pixels between grid points linearly interpolated. Activity overlapping pixels within intact or manipulated faces were averaged and plotted as post stimulus time histograms (smoothed using a gaussian kernel over 20 ms sliding window). This activity was compared with the mean activity over pixels of other parts of the image. Significance was assessed across images with paired, two-way t-tests, corrected for a false discovery rate adjusted p value of 0.05.

## Convolution neural network image analysis

240 greyscale images evenly distributed across 6 categories (human faces, headless human bodies, animals, houses, tools, and inanimate objects) were analyzed. Each category image was presented on a uniform white background. Images were analyzed using the MATLAB implementation of a pre-

trained convolutional neural network, AlexNet. Network activations across relu layers of AlexNet were computed for each image. To measure the representational similarity across images within a layer, Spearman rank correlations were computed between z-scored activations for each image pair. To visualize the representational structure of each layer, multidimensional scaling was performed on squared Euclidean distances between the z-scored activation patterns of image pairs. To probe distinctions purely based on image shape, correlations and multidimensional scaling were also performed on the vectorized 3 color-channel pixel intensities for each image. Permutation testing was performed by randomizing the labels corresponding of non-face images prior to computing correlations between face and non-face categories. Mean correlations and the 2.5% and 97.5% percentiles across 2500 randomizations were computed for each layer of AlexNet and for pixel intensity. To look at the effect of training, an additional permutation test was performed where the weights within each layer were shuffled prior to measuring the activations for each image and computing image pair correlations. This approach simulates an untrained network while preserving the distribution of weights found in the trained network. The mean correlations and the 2.5% and 97.5% percentiles across 500 permutations were computed for each layer of AlexNet.

### Quantification and statistical analyses

For activity maps in *Figures 1–7*, *Figure 1—figure supplements 2*, *4* and *5*, *Figure 2—figure supplement 1*, *Figure 3—figure supplements 1–3*, *Figure 4—figure supplements 1–3*, *Figure 5—figure supplements 1–3*, *Figure 6—figure supplements 1–3*, and *Figure 7—figure supplement 2* spike data were averaged across trial repetitions. For PSTH graphs in *Figures 1–7* and *Figure 4—figure supplement 4*, the mean response and 95% confidence intervals across images were calculated. For each 1 ms bin, two-sample, paired t-tests were performed across images on the firing rates between the target and control regions. The resulting p values were adjusted for false discovery rates. Given, the relatively small number of observations going into each t-paired t-test, significant effects were verified further with nonparametric (Wilcoxon signed-rank) tests.

## Acknowledgements

We thank A Schapiro for advice on analyses and helpful comments on the manuscript. This work was supported by NIH grants RO1 EY 16187 and P30 EY 12196.

## Additional information

### Funding

| Funder | Grant reference number | Author |
| --- | --- | --- |
| National Institutes of Health | RO1 EY 16187 | Margaret Livingstone |
| National Institutes of Health | P30 EY 12196 | Margaret Livingstone |

The funders had no role in study design, data collection and interpretation, or the decision to submit the work for publication.

### Author contributions

Michael J Arcaro, Conceptualization, Resources, Data curation, Software, Formal analysis, Validation, Investigation, Visualization, Methodology, Writing - original draft, Writing - review and editing; Carlos Ponce, Conceptualization, Software, Visualization, Methodology; Margaret Livingstone, Conceptualization, Resources, Data curation, Formal analysis, Supervision, Funding acquisition, Investigation, Visualization, Writing - original draft, Writing - review and editing

### Author ORCIDs

Michael J Arcaro (iD) https://orcid.org/0000-0002-4612-9921

## Ethics

Animal experimentation: All procedures were approved in protocol (1146) by the Harvard Medical School Institutional Animal Care and Use Committee (IACUC), following the Guide for the care and use of laboratory animals (Ed 8). This paper conforms to the ARRIVE Guidelines checklist.

## Decision letter and Author response

Decision letter https://doi.org/10.7554/eLife.53798.sa1
Author response https://doi.org/10.7554/eLife.53798.sa2

## Additional files

### Supplementary files

• Transparent reporting form

### Data availability

Custom scripts and data to reproduce figures have been deposited to GitHub (https://github.com/mikearcaro/NoHeadResponseMaps; copy archived at https://github.com/elifesciences-publications/NoHeadResponseMaps). The complete dataset that was collected is available upon reasonable request to the corresponding author.

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
