## [Decision Letter]

**Acceptance summary:**

The finding that the representation of faces in inferotemporal cortex depends on the location of the animal's fixation and is context specific was judged as novel and exciting. The elegant and unique design of the study allowed the authors to show that face-selective neurons respond not just to faces in isolation but also to parts of an image that do not contain a face but should do so, based on the context (proximity to the body). The work provided convincing evidence that the representation of complex objects in the brain reflects statistical regularities of the experienced environment.

**Decision letter after peer review:**

Thank you for submitting your article "The neurons that mistook a hat for a face" for consideration by *eLife*. Your article has been reviewed by two peer reviewers, and the evaluation has been overseen by a Reviewing Editor and Timothy Behrens as the Senior Editor. The reviewers have opted to remain anonymous.

The reviewers have discussed the reviews with one another and the Reviewing Editor has drafted this decision to help you prepare a revised submission.

Summary:

The authors report a series of recording experiments in macaque face patches (PL and ML) that examined the response of face responsive neuronal sites to natural images that contain bodies but with the faces occluded or obscured. Although the faces were absent or only partially visible, the face patch ML neurons tended to respond at the image location where the face was supposed to be. They conclude that face neurons of the face patch ML are sensitive to contextual influences and experienced environmental regularities. This paper reports novel data on the response properties of face neurons in face patch ML: responses to the location of a face when the latter is absent or occluded by another object but suggested by the context of a body. The experimental paradigm is also quite interesting and unique in this field: presenting a complex natural image at different locations and then analyzing the response as a function of the location or hotspot of the receptive field of the neuron as measured with a single face stimulus.

Essential revisions:

The manuscript was well received by both reviewers who found the work exciting and the results novel. However, they raised a number of issues, mainly concerned with the need for more detailed description of the data to give the readers a more rigorous impression of the observed effects. To achieve this goal, the paper needs better quantification of the prevalence of the effect and its presence in individual neurons in the recorded population. To help the authors, the reviewers provided a detailed list of questions, requests and suggestions, which must be addressed before the manuscript will be consider for publication in *eLife*. These are summarized below.

*Reviewer 1:*

1) Please clarify the data in Figures 2 and 3 by providing cell numbers for each plot and for each recorded region. In addition, provide information about stimulus selectivity of the recorded neurons and discuss the relationship between this selectivity and responses to faces shown in these figures.

2) Were the RFs showing responses to images in Figure 1—figure supplement 1, also mapped with small dots?

3) Please clarify the composite body orientation maps shown in Figure 6.

*Reviewer 2:*

1) Please discuss some of the discrepancies between the data and their description in the text outlined in detail by this reviewer. Also, please address the question whether scaling maps to the minimum and maximum response for each image prevents detecting responses at other locations. Also, in Figures 5 and 6B please show maps for the images, in addition to the PSTHs.

2) Differences in activity between different recording sites need to be discussed. A related question is the definition of non-face regions. This reviewer makes specific suggestions concerning Figure S6, which you may find helpful in dealing with the problem.

3) Please discuss responses to the legs for inverted bodies shown in Figure 6 and whether these multi-unit responses reflect mixed selectivity with the recorded patch. Showing responses of well-isolated units would help addressing this possibility.

4) Long latencies of responses to occluded head images cast doubt on the interpretation proposed in the Discussion that rules out top-down explanation of the effect. In addition, the proposed role of experience is hypothetical and should be presented as such.

5) This reviewer also made several points that should be addressed in the body of the paper or in the Discussion. These include, clarifications of mapping RFs with dots and the possibility that their size was underestimated, dealing with negative response values when computing the index, enlarging the size of Figures 4 and S6 for better visibility and correction of the legend for Figure 4.

6) Please comment on the validity of using parametric t-tests to evaluate the effects and whether the assumptions underlying these tests are fulfilled.

*Reviewer #1:*

This study demonstrates that neurons in two macaque face patches, ML and PL, respond in a retinopically specific way to face-related features of a scene. Surprisingly, these face-related responses are often elicited even when the basic structural features of a face are occluded, such that only the bodily cues indirectly indicate the presence of a face in the neuron's response field.

1) This is a very exciting study that has makes two important points. The first, which is not featured, perhaps because it is "sort of" known, is that for neurons in these face patches the animal's fixation position is extremely important. Responses to faces or "unseen" faces are wholly contingent on the face being present on a limited portion of the retina. I thought this aspect of the study is very important for a community that does not typically think about "face cells" in this way. The second aspect, which is featured more strongly, is that neurons respond in this retinotopically specific way to occluded faces. This quite a remarkable finding. The authors do some interesting extensions and controls, including showing a surprising lack of influence by adding the body to the visible face. This study provides much food for thought.

2) My main criticism of the paper is that I don't have a clear picture of how representative the featured effect is. Figures 2 and 3 show heat maps, and refer to "maps of cells". Are these population maps (as in the bottom panels of Figure 1—figure supplement 3), or are they selected responses of individual cells? There is a statement early on that everything is population analysis, but the authors need to do a better job explaining how many cells go into each of the plots, since the reader needs to understand to what extent this behavior is typical of neurons in ML and PL. Having only one PL recording site also significantly weakens this aspect.

3) Relatedly, since neurons differ in their specific preferences, it's surprising that there is minimal discussion of this. The authors should do more to provide information about how stimulus selectivity (aside from positional selectivity) interacts with their main findings.

*Reviewer #2:*

The authors report a series of recording experiments in macaque face patches (PL and ML) that examined the response of face responsive neuronal sites to natural images that contain bodies but with the faces occluded or obscured. Although the faces were absent or only partially visible, the face patch ML neurons tended to respond at the image location where the face was supposed to be. They conclude that face neurons of the face patch ML are sensitive to contextual influences and experienced environmental regularities.

This paper reports novel data on the response properties of face neurons in face patch ML: responses to the location of a face when the latter is absent or occluded by another object but suggested by the context of a body. The experimental paradigm is also quite interesting and unique in this field: presenting a complex natural image at different locations and then analyzing the response as a function of the location or hotspot of the receptive field of the neuron as measured with a single face stimulus. The main result agrees with previous fMRI studies in human and monkeys that showed responses in face areas / patches to blurred faces on top of a body, but the present study examines this in some more detail at the spiking activity level in the posterior face patches.

In general, I found this a highly interesting and exciting paper but I have some concerns and comments that need to be addressed, in particular with respect to the presentation and interpretation of these remarkable data.

1) The results are not as clear-cut as the impression one gets from reading the text of the paper. Examples of such issues are the following. Figure 2: Monkey 1 ML: second column image (human carrying a box): in the occluded condition there is also strong activation in the top right corner where the box is. In Figure 3, third and fifth column: the location of the masked face and the hotspot of neural activity do not match well in the data of M2. Furthermore, for the 7th column image of Figure 3 (3 persons carrying boxes) there is a clear hotspot in both monkeys only for the rightmost (third) person: why not for the closer other two persons? In the 9th column image, there is an additional hotspot in the data of M2 on the body below the covered head. Also striking is the variability in effects between images and monkeys. This is apparent in the data of Figure 4: second column (little response for ball covering the head in M1), fourth column (no or little response to non-face object on face), fifth column (no response to covered face in M1). Also, in general the data of M2 appear to be more "smeared out" compared to those of M1: is that related to RF size differences, neuron-to-neuron variability or the mapping with larger steps in M2? The authors should discuss these discrepancies between the data and what they describe in the text in detail, acknowledging that these cells do not always respond at the location of the occluded head or sometimes even not close to the occluded head. One related problem concerning the data presentation is that the maps are scaled to the minimum and maximum response for each image. I wonder how the maps will look like when shown across the full possible range of net responses for a site (i.e. from 0 net response to maximum response): was there no response at other locations than (or close to) the (occluded) head/face? Also, related to this point, the authors should show the maps for the images of Figure 5 and 6B – not just the PSTHs (see next main point). In other words, these interesting data should be presented with greater detail.

2) The authors show only averages across electrodes per animal (except for the responses to unoccluded faces (Figure S3)) and one wonders how strong the differences are between the different neuron sites. To remedy this, the authors can show the maps for each image in Figure S6 (but please make the images larger) so that the reader can assess the (or lack of) consistency of the responses for the different images. The PSTHs in the current figures are averages across the images so the presented confidence intervals give some indication of the variability for the different images. However, what was unclear to me is how the authors defined the non-face region? They should indicate the latter in Figure S6 and describe in detail how they define the non-face region: centered on an object, equated for contrast/luminance with the face/occluded face, same area, etc? Choosing a valid non-face region does not seem trivial to me for these complex images, and such a choice may affect the responses shown in the PSTHs. Finally, the authors can compute for each site and image an index contrasting the net response to the face and control, non-face region and show the distribution of such indices for each experiment.

3) In the experiment of Figure 5, the authors compare the responses to a modified face alone and the modified face on top of a body and find stronger responses when the body is present. However, they should have as a control condition also the body without face, otherwise the increased response could have been due to a (weak) response to the body itself (summation of obscured face and body response). The absence of such summation effect for full faces could be due to a ceiling effect, as suggested by the authors.

4) The strong responses to the legs for the inverted body in Figure 6 is surprising to me. It clearly shows that the responses of these neurons to body stimuli without a head are more complex than just a response to an occluded face or absent head. The authors suggest that this is related to responses to short contours in the face cells. But if that is the case, can it not explain also the responses to the rectangular shapes (with also short contours) occluding the head in the images of Figure 3? Also, when it is related to short contours, why not then comparable strong responses to the images with a head (upper row of Figure 6)? It is clear to me that we do not understand the responses to these neurons. Also, how consistent was this leg response across the different sites? It also makes one wonder whether some of these "weird" responses are because the recordings are multi-unit, which will pool the responses of different neurons, some of which may not be face selective. In other words, it could be that the single neurons that respond to the legs are not face selective at all but mixed with face selective neurons in this patch. It would have been informative to show the responses of well-isolated single units in this case (and also in the other experiments).

5) The authors suggest in the Discussion that the responses to the occluded head are not due merely to expectation. However, the latencies of the responses to the occluded head appear to be longer than to a real face (and, in some cases, the responses were (as for the non-face) preceded by inhibition (Figure 2, 3, 4 and 5 in M1)). Thus I feel one cannot rule out an expectation or top-down explanation of the effects.

6) That the responses to the occluded heads result from experience makes sense but is speculation and not demonstrated in the present experiments. This should be made explicit.

[Editors' note: further revisions were suggested prior to acceptance, as described below.]

Thank you for resubmitting your article "The neurons that mistook a hat for a face" for consideration by *eLife*. Your article has been reviewed by two peer reviewers, and the evaluation has been overseen by a Reviewing Editor and Timothy Behrens as the Senior Editor. The following individual involved in review of your submission has agreed to reveal their identity: David Leopold (Reviewer #1).

The reviewers have discussed the reviews with one another and the Reviewing Editor has drafted this decision to help you prepare a revised submission.

Summary:

The revised manuscript addressed many of the issues raised in the initial review. However, one of the reviewers raised additional concerns and made several suggestions that must be addressed for the manuscript to be accepted for publication. These are listed below.

Essential revisions:

1) Please provide the distribution of indices to allow the reader better assess the variability of the data.

2) Please create an additional figure that compares directly PSTHs recorded in the ML and PL face patches (see reviewer 2 comment #3)

3) The claim that the changes in face-body representations at successive layers of Alexnet simulations result from learning, requires additional analysis of category representations in an untrained Alexnet using random weights.

4) Please tone down the interpretation of the importance of the eyes when referring to Figure 2 in the second paragraph of the Results section.

*Reviewer #1:*

The authors have done an outstanding job addressing my comments, clarifying a few bits and then showing many single-cell examples of this striking phenomenon in the supplementary material. There are many important aspects of this paper beyond the headline, including the spatial specificity of face selective neurons, as well as the similar but delayed responses in ML compared to PL. I'm happy to recommend publication.

*Reviewer #2:*

The authors performed an extensive revision, with new interesting data and modelling. They responded to all my concerns and overall I am happy with their revision. Nonetheless, I still have a couple of concerns that should be addressed by the authors.

1) I understand that it is not possible to show the responses of all units to all stimuli, given the size of the data. I asked to show distribution of indices in order to get an idea about the across image and unit variability but the authors argue that this is not practical because of the variability in response latency amongst images and units. Such argument can always be made and then one should refrain from using indices (but the authors compute face and body indices). I think that when one uses net responses to compute indices, the variability in latency should not be a main cause of variance among indices of different neurons or images. One can employ an analysis window that captures the population response, computed across images and neurons, per experiment. The confidence intervals that are provided now depend on the number of observations; although they show the consistency of the mean population response, the CIs are not a straightforward metric of the variability among sites or images. Also, it is not clear whether the individual site examples shown in the figure supplements are representative of the observed diversity of response profiles. Hence, I still believe that distributions of indices will help to appreciate the variability among images and sites.

2) The authors provide novel data on the responses to images in which the eye is invisible. However, in the images shown in Figure 2, either more than the eyes are removed (top image) and it is unclear how these other changes affected the response to the disfigured face, or, the eyes are occluded by a visible occluder (bottom image). The latter is known to increase the response latency of IT neurons. To assess whether removing the eye from the image affects the response, it would have been better to present a face without eyes, the eye region being filled in with the neighboring texture, so that no additional changes (holes or bar occluder) are present. With the current images, I believe the authors should be more careful in interpreting the response and latency differences between the full face and the images in which the eyes were removed or occluded.

3) The authors suggest that the responses to the occluded faces do not result from top-down processing, using as argument that the response latency differences between ML and PL remain for the occluded faces, which is a reasonable argument. However, PL and ML responses are compared directly only for the intact faces and the "invisible eye" images, but not for the face occlusion conditions. One way to show this is to compare in one figure the population PSTHs between ML and PL for the different conditions.

4) They write, when describing the novel and interesting Alexnet simulations, that "The Euclidean distances between image pairs within each category were smaller than between categories, indicating that images were more homogeneous within than between categories and highlighting that simple shape is a major component of these commonly tested "semantic" categories". However, the images also differ in texture and it is possible that not only shape but also texture is driving the category differences in Alexnet. In fact, some recent studies stressed the importance of texture cues for categorization in deep convolutional neural networks.

5) The authors employ the Alexnet simulation to demonstrate that representations of faces and bodies can merge over learning. However, to show that learning is critical they should perform a similar analysis of the category representations in an untrained Alexnet, e.g. with random weights.

6) The authors write "that convolutional networks can capture the representational structure of biological networks." It all depends on what is meant by "capture" here, but there is still a discrepancy between IT and classical feedforward deep net (like Alexnet) representations, e.g. see recent work by DiCarlo's group. I suggest to tone down this statement on the correspondence between deep nets and ventral stream representations.

7) Can the authors explain why they permute the labels of the non-face images when computing the between face, non-face correlations (inset; Figure 8B); permutation of the non-face labels in this case should not make a difference, or perhaps I am missing something here.

8) The authors should provide the explained variance of the 2D MDS configuration shown in their novel Figure 8A, so that the reader can appreciate how the distances plotted in 2D correspond to the original distances in the high-dimensional feature space.

9) The authors write in their Discussion "More broadly, these results highlight.… that the facilitation of face-selective neurons in the absence of a clear face reflects a lifetime of experience that bodies are usually accompanied by faces.". As I noted before, there is no evidence yet that the contextual effects reported by the authors do result from experience during the lifetime. Please rephrase.

10) Figure 1—figure supplement 1: the authors should indicate the t-values (with the corrected p = 0.05 threshold) of the face-object contrast in the fMRI maps.

11) Figure 1—figure supplement 3: can the authors explain the above-baseline responses in M1, ML before 50 ms after stimulus onset – one would expect that soon after stimulus onset responses equal to baseline because of the response latency of the neurons.

---

## [Author Response]

Essential revisions:The manuscript was well received by both reviewers who found the work exciting and the results novel. However, they raised a number of issues, mainly concerned with the need for more detailed description of the data to give the readers a more rigorous impression of the observed effects. To achieve this goal, the paper needs better quantification of the prevalence of the effect and its presence in individual neurons in the recorded population. To help the authors, the reviewers provided a detailed list of questions, requests and suggestions, which must be addressed before the manuscript will be consider for publication in eLife. These are summarized below.

We thank the reviewers for their thorough reviews and positive assessment. We provide a point-by-point response to each of the reviewers’ comments. In summary, we made the following revisions to our manuscript:

1) We have made extensive edits to the text to address reviewer comments.

2) We provide additional quantifications of response consistencies and statistical analyses as requested.

3) We have added 12 additional supplementary figures (Figure 3—figure supplements 1-3, Figure 4—figure supplements 1-3, Figure 5—figure supplements 1-3, Figure 6—figure supplements 1-2, and Figure 7—figure supplement 2) that show more individual channel and population-level responses maps to images reported in the main text figures.

4) Using data we previously collected, but had not reported, we have added 1 new main text figure (Figure 2) and 3 additional supplementary figures (Figure 1—figure supplement 6, Figure 2—figure supplement 1, and Figure 6—figure supplement 3) that show response maps to additional images that illustrate the selectivity and latency of responses.

5) We have added a new analysis of category representations using an artificial neural network (Figure 8).

6) We have added a new supplementary figure illustrating the localization of the arrays to the face patches in IT.

Reviewer 1:1) Please clarify the data in Figures 2 and 3 by providing cell numbers for each plot and for each recorded region. In addition, provide information about stimulus selectivity of the recorded neurons and discuss the relationship between this selectivity and responses to faces shown in these figures.

Data in Figures 3 and 4 (previously Figures 2 and 3) are population activity (averaged across all responsive channels within an array). We have clarified this in the figure legends as well as in Figures 5 and 7. Previously, we showed individual channel data from Monkey 1 and Monkey 2 only for Figure 1 in Figure 1—figure supplement 4. Now, we include supplementary figures that show 3 individual channels each for Monkey 1 and Monkey 2 for the maps in Figure 3 (Figure 3—figure supplement 3), Figure 4 (Figure 4—figure supplement 3), and Figure 5 (Figure 5—figure supplement 3). To better convey the consistency of responses across channels, we now also report the average Pearson correlation between each channel and the n-1 population average (i.e. leaving out that channel) for data in Figures 3 – 5.

2) Were the RFs showing responses to images in Figure 1—figure supplement 1, also mapped with small dots?

We apologize for this confusion. That was a mistake in the text. The RFs in Figure 1—figure supplement 2 (formerly Figure 1—figure supplement 1) were mapped with 2 degree faces and round non-face images. We now show examples of the faces and round non-face images in Figure 1—figure supplement 2 and report the mean and standard deviations of RF sizes across channels for each array in the legend. In a separate mapping session (not reported in manuscript) we did perform RF mapping in Monkey M1 using face features such as eyes and simplified versions of eyes (small dots and small concentric circles). Similar to the mapping with 2º faces and objects, the centers of those RFs were within the central visual field, but the RFs were even smaller (mean RF 1.41 +/- 0.24 STD for Monkey 1 ML). The RF centers were in similar locations using these stimuli. We did not perform this mapping with face features in monkey M2. For the purposes of our complex natural image recordings, only the center of the RF is critical for the analysis. If anything, larger RFs would reduce the resolution of focal activity and minimize the effects observed in our main experiments. The fact that activity was focal to faces in natural images demonstrates that the RFs were not so large as to be a major constraint on our results.

3) Please clarify the composite body orientation maps shown in Figure 6.

The composite orientation maps serve as summary data illustrating the best body orientation at each grid point. We feel this is important to include given the variability in response magnitudes across both body orientations and spatial location. This is especially true for upright vs. inverted headless bodies as shown more clearly in Figure 7B. Because of this, one cannot look at any individual orientation to know what the preferred tuning is for that point in space. As requested by the reviewers, we have made the color scale and corresponding body orientation images on the right side of Figure 7A (formerly Figure 6) larger to improve readability. We also changed the colormap for the composite orientation images to make them more distinct from the response maps so readers would be less likely to confuse those data for firing rates. We have included the following explanation of these composite maps in the Results:

“The magnitude of firing rate varied across the tested visual field location (e.g., firing rates tended to be higher above the upper half of the image). Composite maps show the preferred body orientation for each spatial position tested (Figure 7A, right side). In experiments both with and without heads, preferred orientation systematically varied such that the preferred body orientation at that spatial location positioned the face, or where the face ought to be, within the RF center.”

Reviewer 2:1) Please discuss some of the discrepancies between the data and their description in the text outlined in detail by this reviewer.

Reviewer #2 made several comments noting the variability of the activity maps and appearance of responses to natural scenes that are not clearly driven by the body. We address this above in main comment #1 as well as below with reviewer #2’s specific comments.

Also, please address the question whether scaling maps to the minimum and maximum response for each image prevents detecting responses at other locations.

In our initial submission we chose a min to max response scaling to illustrate the full range of responses. However, large outliers can bias the appearance of response maps scaled between minimum and maximum values. To minimize the influence of outliers and to better center the color scale range to the values for each map, we now show every response map scaled between the 0.5 and 99.5% percentiles of each map. This has the effect of increasing the effective range for the bulk of each response map. We do not think this affects the interpretation of the response maps. We also looked at using 1% to 99% and 2.5 to 97.5% ranges, but found that 0.5% to 99.5% best illustrated the range of values within activity maps. To clarify, the minimum response for Monkey 1 is often slightly below baseline as can be seen in the PSTH responses to non-face regions. For Monkey 2, the minimum response was close to 0 with some images slightly below and others slightly above.

As the reviewer noted above, for any individual image, there are sometimes hot spots (yellow / red) outside the face area (or where the face ought to be). We are not claiming that these neurons respond only to the face stimuli; see for example Tsao et al., 2006 showing that face cells consistently respond to clocks and round fruits. Indeed, we also see consistent responses particularly to round, tan objects such as cookies. See Author response image 1. It seems likely that at least some of these features drive these neurons because they share shape, color, or contrast features with faces. There may be other features that consistently drive these neurons across images. Relatively few studies have systematically tested tuning of face patch neurons to features embedded within naturalistic stimuli. This is certainly an interesting avenue of exploration, but beyond the scope of the current study. Here, we specifically study the effect of bodies positioned below face selective neuron RFs on firing rate

**Author response image 1. respfig1:** Response maps from M1 to natural images without faces. Hotspots are consistently found over circular shapes and/or above body-life structures.

Also, in Figures 5 and 6B please show maps for the images, in addition to the PSTHs.

We now show response maps for each stimulus in Figures 6 and 7B (previously Figures 5 and 6B). See Figure 6—figure supplements 1 and 2 and Figure 7—figure supplement 2.

2) Differences in activity between different recording sites need to be discussed. A related question is the definition of non-face regions. This reviewer makes specific suggestions concerning Figure S6, which you may find helpful in dealing with the problem.

The response selectivity across the arrays in both animals was consistent. In Figure 1—figure supplement 4, we provided a measure of the similarity (Pearson correlations) between each channel and the n-1 population average for responses to natural images with faces. We now provide examples of individual channels for the population-level response maps shown in Figure 3 (Figure 3—figure supplement 3), Figure 4 (Figure 4—figure supplement 3), and Figure 5 (Figure 5—figure supplement 3). To better convey the consistency of responses across channels, we now also report the average Pearson correlation between each channel and the n-1 population average for data in Figures 3– 5.

We did not see mixed selectivity in any of the arrays. We now provide indices for bodies vs. objects and faces vs. all other images for each responsive channel in all 3 recording sites in Figure 1—figure supplement 3. We find little to no body selectivity. For the experiments reported in Figures 3-5, we also now provide the mean correlation and variance between the spatial pattern of response maps for each channel and the n-1 group average within the ML arrays.

3) Please discuss responses to the legs for inverted bodies shown in Figure 6 and whether these multi-unit responses reflect mixed selectivity with the recorded patch. Showing responses of well-isolated units would help addressing this possibility.

We show in Figure 1—figure supplement 3 that these neurons responded to faces, but did not respond to bodies, when the bodies were presented within the receptive field. We find no evidence of mixed selectivity in our multi-unit RSVP data (e.g., strong body or hand selectivity). Reviewer #2 raised the concern that our multi-unit data comprises a predominant pool of face-selective neurons and a weaker pool of body-selective neurons. Such an interpretation is not consistent with our results. First, images of bodies did not significantly activate these neurons when presented within the RF (Figure 1—figure supplement 3). We now provide body-selectivity indices showing that there was little to no body selectivity in these arrays. Across our experiments, hot spots of activity are consistently above the body, not on it. A mixed-selectivity explanation would need the RFs of face-selective neurons to be systematically right above the RFs of body-selective neurons in both monkeys. However, such an account is not consistent with our results reported in Figure 6 (also see new Figure 6—figure supplements 1 and 2 for response maps for every image in this experiment). We show that the response to an intact face alone or above a body is near identical in magnitude. If the face and body responses were separate neuronal populations, these responses would have to summate and the face + body response would need to be larger than the face without body. Therefore, the only parsimonious explanation is that our multi-unit data comprised face-selective neurons that respond to a face presented within the RF or to a body presented below the RF. Upon consideration of the reviewers’ comments, we agree that the link to end-stopping was to speculative. In the results, we now limit our interpretation that the response to feet for inverted headless bodies may be a manifestation of this body-below responsiveness.

4) Long latencies of responses to occluded head images cast doubt on the interpretation proposed in the Discussion that rules out top-down explanation of the effect. In addition, the proposed role of experience is hypothetical and should be presented as such.

We understand the reviewer’s concern regarding longer latencies. We do not agree that long latencies cast doubt on our interpretation, nor that it necessarily suggests a top-down explanation. In fact, the latencies of these neurons can vary simply by changing the size of a face or adding/removing backgrounds. To better illustrate the dependency of latencies on the visual input, we now include two new figures (Figure 2 and Figure 2—figure supplement 1) showing the response latencies of PL and ML simultaneously recorded in Monkey 1 to natural face images with and without eyes. By simply removing the eyes from faces, we see a substantial shift in the latencies for both PL and ML. The ML response latencies for these images without eyes is comparable to the latencies of the masked and occluded faces. Importantly, PL still responds to the eyeless faces with a shorter latency than ML. We do not think this suggests top-down modulation, rather that these neurons are sensitive to the input of a variety of visual features that can manifest at different timescales. For example, we used a stimulus with parts of a face spatially separated (top of Figure 2). For both PL and ML, the response to eyes emerged in time windows earlier than the response to the surrounding face. This is consistent with prior findings in PL by Issa and DiCarlo (JN 2012; Figure 12).

In the discussion, we propose that contextual representations such as a body below a face can manifest locally within IT through experience. We do not claim that we have shown that experience is necessary but think this is quite plausible. To support this interpretation, we now provide an analysis of category representations using an artificial neural network trained on image classification (Figure 8). We show that representations of faces and bodies in an artificial neural network (AlexNet) trained to classify images into a set of 1000 categories (but not explicitly trained on face, body, or human categorization!) start off separate and merge closer to one another in deep layers of the architecture more than with other categories. We show this with multidimensional scaling visualizations of the distances between images for activations in early and deep layers of AlexNet (Figure 8A). We also show that the overlap in face and body representations is greatest for deep layers (relu6 and relu7) of AlexNet (Figure 8B). This provides some evidence that the regularity of experiencing faces and bodies is sufficient for overlapping representations to emerge in a hierarchical system.

5) This reviewer also made several minor points that should be addressed in the body of the paper or in the Discussion. These include, clarifications of mapping RFs with dots and the possibility that their size was underestimated, dealing with negative response values when computing the index, enlarging the size of Figures 4 and S6 for better visibility and correction of the legend for Figure 4.

We have clarified the issue of RF mapping above and addressed the concern about RF size.

There were no channels with negative (below baseline) responses to faces or non-face objects in the PL array in Monkey 1. There were no negative face or non-face object responses in ML of either monkey but one channel in each of the ML arrays had a below-baseline response to bodies, so those channels were not used in calculating the average body vs. object indices. We clarify this in the results and Figure 1—figure supplement 3’s legend.

6) Please comment on the validity of using parametric t-tests to evaluate the effects and whether the assumptions underlying these tests are fulfilled.

We acknowledge that a relatively small number of observations (images) went into each test, which can be problematic for parametric tests. To alleviate the reviewer’s concerns, we re-ran all the statistical analyses using non-parametric Wilcoxon tests. The results were nearly identical and did not change any interpretation of the results. We provide in Author response image 2 an illustration of the parametric vs. non-parametric results (both threshold at p < 0.05 FDR-corrected) for all PSTH graphs with significant results.

**Author response image 2. respfig2:** Comparison of t-test results with non-parametric Wilcoxon signed-rank test. Both threshold at p< 0.05 FDR-corrected.

Reviewer #1:This study demonstrates that neurons in two macaque face patches, ML and PL, respond in a retinopically specific way to face-related features of a scene. Surprisingly, these face-related responses are often elicited even when the basic structural features of a face are occluded, such that only the bodily cues indirectly indicate the presence of a face in the neuron's response field.1) This is a very exciting study that has makes two important points. The first, which is not featured, perhaps because it is "sort of" known, is that for neurons in these face patches the animal's fixation position is extremely important. Responses to faces or "unseen" faces are wholly contingent on the face being present on a limited portion of the retina. I thought this aspect of the study is very important for a community that does not typically think about "face cells" in this way. The second aspect, which is featured more strongly, is that neurons respond in this retinotopically specific way to occluded faces. This quite a remarkable finding. The authors do some interesting extensions and controls, including showing a surprising lack of influence by adding the body to the visible face. This study provides much food for thought.

We thank the reviewer for this encouraging assessment.

2) My main criticism of the paper is that I don't have a clear picture of how representative the featured effect is. Figures 2 and 3 show heat maps, and refer to "maps of cells". Are these population maps (as in the bottom panels of Figure 1—figure supplement 3), or are they selected responses of individual cells? There is a statement early on that everything is population analysis, but the authors need to do a better job explaining how many cells go into each of the plots, since the reader needs to understand to what extent this behavior is typical of neurons in ML and PL. Having only one PL recording site also significantly weakens this aspect.

We apologize for this confusion. All main text figures show population data – all visually responsive sites in each array. Figure 1—figure supplement 4 showed a few examples of response maps from a single channel for comparison with the population map for Monkeys 1 and 2. We focus on the population data because the selectivity from a RSVP paradigm (Figure 1—figure supplement 3) was similar across channels within each array and the response maps of individual channels (on average) to complex images containing faces were highly similar to the n-1 group average, so therefore the population response maps were representative of consistent activity across individual channels.

To give a better sense of the individual channel variability, we now include graphs of body- and face-selective indices for each array (Figure 1—figure supplement 3) and include new supplemental figures for Figures 3-5 (previously Figures 2-4) showing response maps for 3 individual channels each for both Monkeys 1 and 2. We also now report the similarity of response maps across channels within each array for the experiments in Figures 3-5.

We agree that only having a single PL recording array from one monkey limits what we can conclude about PL in general. Overall face-selectivity and the responses to occluded faces were weaker in PL so we focused on ML. We still feel that it is useful to report our PL results here, given that in this monkey we recorded simultaneously from PL and ML (which to our knowledge has not been reported previously). To make better use of the PL data, we now include a figure illustrating the latency differences between PL and ML from simultaneous recordings. For each natural image, the face response to PL preceded ML. Interestingly, when we occluded the eyes, the latencies for both PL and ML increased, but PL face responses still preceded ML. Though PL showed only weak responses to masked and occluded faces, we still saw a similar latency difference (Figure 2—figure supplement 2). We take this as additional evidence that the body-facilitation effect is not due to top-down feedback. Implanting a PL array in Monkey 2 is not an option at this time, so adding another PL monkey now would require training up a third monkey, targeting, and implanting PL, which would take months.

3) Relatedly, since neurons differ in their specific preferences, it's surprising that there is minimal discussion of this. The authors should do more to provide information about how stimulus selectivity (aside from positional selectivity) interacts with their main findings.

We did not see much heterogeneity in tuning across each of our arrays. We attribute this homogeneity in each array to targeting face patches using fMRI. In Figure 1—figure supplement 3, we show, using an independent stimulus set of isolated objects, that all visually responsive channels selectively responded to faces (as compared to bodies, hands, and objects). We now provide body- and face- selectivity indices for each channel. The degree of face selectivity varied across channels, but all channels were much more responsive to faces than to any other category. These neurons may vary in their preference for particular facial features. We did not test for this though all channels were particularly responsive to eyes. There was some variability in the specific tuning e.g., some face-selective channels in Monkey 2 had a small preference for human faces over monkey faces, and other channels in Monkey 2 preferred the opposite. We did not thoroughly test other dimensions that face cells are known to vary along such as viewpoint preference, and our complex stimuli did not substantially vary along this dimension. To better illustrate the similarity in response selectivity across channels, we now show response maps for 3 individual channels for Figures 3 – 5 (previously Figures 2 – 4).

Reviewer #2:[…]1) The results are not as clear-cut as the impression one gets from reading the text of the paper. Examples of such issues are the following. Figure 2: Monkey 1 ML: second column image (human carrying a box): in the occluded condition there is also strong activation in the top right corner where the box is. In Figure 3, third and fifth column: the location of the masked face and the hotspot of neural activity do not match well in the data of M2. Furthermore, for the 7th column image of Figure 3 (3 persons carrying boxes) there is a clear hotspot in both monkeys only for the rightmost (third) person: why not for the closer other two persons? In the 9th column image, there is an additional hotspot in the data of M2 on the body below the covered head. Also striking is the variability in effects between images and monkeys. This is apparent in the data of Figure 4: second column (little response for ball covering the head in M1), fourth column (no or little response to non-face object on face), fifth column (no response to covered face in M1). Also, in general the data of M2 appear to be more "smeared out" compared to those of M1: is that related to RF size differences, neuron-to-neuron variability or the mapping with larger steps in M2? The authors should discuss these discrepancies between the data and what they describe in the text in detail, acknowledging that these cells do not always respond at the location of the occluded head or sometimes even not close to the occluded head.

The reviewer is correct that responses were variable across individual stimuli. That is to be expected from recordings in IT cortex. Critically, as we show for each and every experiment, across stimuli responses to faces, or where the face ought to be, are on average stronger than to non-face parts of these images. For each experiment, we show averaged PSTHs along with confidence intervals showing the variability across images and perform statistical testing to convey the reliability of these responses.

As the reviewer noted, it is interesting that for images with multiple faces, some hot spots were more prominent than others. We have been testing the effect of clutter on face responses more directly in a separate set of experiments, but this is beyond the scope of the current study.

Regarding Monkey 2’s activity maps. We note in the Materials and methods and in describing the data reported in Figure 1 that the step size used for mapping Monkey 2’s response was double that of Monkey 1. This was a limitation due to the length of time Monkey 2 would work on a given day. Importantly, the spatial spread of the response maps due to this larger step size did not preclude finding statistically significant responses to bodies below the receptive field, as reported for all seven experiments. As shown in Figure 1—figure supplement 4, the response maps were similar across channels in Monkey 2’s array, therefore it is unlikely that channel variability had a major effect of smearing the response maps. The receptive field sizes across Monkey 2’s ML array were similar to Monkey 1’s (Figure 1—figure supplement 2). For large RFs, we would certainly expect more spatial spread. If anything, large RFs would work against/prevent the spatially specific effects we find.

We agree with the reviewer that this variability across individual stimulus response maps should be acknowledged. In each section we now include the following clarification:

For Figure 3:

“The magnitude and timing of responses to occluded faces varied across stimuli. Across all stimuli, the population level activity to occluded faces was reliably higher than the average response to the rest of each image (t-test across 8 images; p < 0.05, FDR-corrected).”

For Figure 4:

“The population-level responses to masked faces varied in magnitude and timing but were consistently stronger than average responses to the rest of each image (t-test across 12 images; p < 0.05, FDR-corrected).”

For Figure 5:

“Similar to the occluded and masked faces, population-level responses to these face-swapped images varied in magnitude and latency but responses to non-face objects were consistently stronger when positioned above a body than when positioned elsewhere (t-test across 8 and 15 images for M1 and M2, respectively; p < 0.05, FDR corrected).”

One related problem concerning the data presentation is that the maps are scaled to the minimum and maximum response for each image. I wonder how the maps will look like when shown across the full possible range of net responses for a site (i.e. from 0 net response to maximum response): was there no response at other locations than (or close to) the (occluded) head/face? Also, related to this point, the authors should show the maps for the images of Figure 5 and 6B – not just the PSTHs (see next main point). In other words, these interesting data should be presented with greater detail.

We address this comment in the Essential Revisions section above.

2) The authors show only averages across electrodes per animal (except for the responses to unoccluded faces (Figure S3)) and one wonders how strong the differences are between the different neuron sites. To remedy this, the authors can show the maps for each image in Figure S6 (but please make the images larger) so that the reader can assess the (or lack of) consistency of the responses for the different images.

We performed statistical analyses for each experiment to quantify the reliability of effects across test images.

Across all experiments we showed 222 images. Each of the 3 arrays recorded 32 channels. We do not think it is reasonable to provide a supplementary figure with 21,312 images (made large). For each experiment we show the mean responses and confidence intervals across all images and provide appropriate statistics to demonstrate reliable responses.

We now provide several additional supplemental figures to illustrate:

1) the variability of the response maps across time (Figure 1—figure supplement 5, Figure 2—figure supplement 1, Figure 3—figure supplement 1, Figure 3—figure supplement 2, Figure 4—figure supplement 1, Figure 4—figure supplement 2, Figure 5—figure supplement 1, Figure 5—figure supplement 2)

2) the variability across channels (Figure 3—figure supplement 3, Figure 4—figure supplement 3, Figure 5—figure supplement 3).

3) population-level response maps for every image reported in Figure 6 (Figure 6—figure supplement 1 and 2) and Figure 7B (Figure 7—figure supplement 2).

The PSTHs in the current figures are averages across the images so the presented confidence intervals give some indication of the variability for the different images. However, what was unclear to me is how the authors defined the non-face region? They should indicate the latter in Figure S6 and describe in detail how they define the non-face region: centered on an object, equated for contrast/luminance with the face/occluded face, same area, etc? Choosing a valid non-face region does not seem trivial to me for these complex images, and such a choice may affect the responses shown in the PSTHs.

For Figures 1, 3 and 4, non-face responses were defined as the mean response across all non-face parts of the image. These regions are indicated as grey in Figure 7—figure supplement 3. We acknowledge that the nonface region averages responses over a wide variety of visual features and choosing a non-face region for comparison is not trivial. Because of this, we performed several additional experiments (reported in Figures 5, 6 and 7B) that provide analyses that have well controlled comparisons. For Figure 5, the response of nonface objects atop a body is compared to responses to the identical objects occurring at another part of the scene. For Figure 6, the PSTHs within each of the 4 face groups (intact, noise, outline, and object) were taken from the same region of the image and comprised identical pixels. The only aspect of the image that varied was the presence or absence of a body below this region of interest. For Figure 7B, the pixels for each image analyzed are identical. The only thing that varies is whether the image was upright or inverted.

Finally, the authors can compute for each site and image an index contrasting the net response to the face and control, non-face region and show the distribution of such indices for each experiment.

The consistency and timecourse of responses to face vs. non-face regions across images in Figures 1-4 is shown in the PSTHs and evaluated statistically. For the mapping experiments, we avoided making such indices because that would require averaging across a set time window, and the latencies were variable across stimuli and experiments. We now provide several additional figures that highlight the variable timecourses of responses (Figure 1—figure supplement 5, Figure 2, Figure 2—figure supplement 1, Figure 3—figure supplement 1, Figure 3—figure supplement 2, Figure 4—figure supplement 1, Figure 4—figure supplement 2, Figure 5—figure supplement 1, Figure 5—figure supplement 2). We feel that a more objective approach is to show the actual responses and provide statistics on the consistency of response across images at each millisecond bin.

3) In the experiment of Figure 5, the authors compare the responses to a modified face alone and the modified face on top of a body and find stronger responses when the body is present. However, they should have as a control condition also the body without face, otherwise the increased response could have been due to a (weak) response to the body itself (summation of obscured face and body response). The absence of such summation effect for full faces could be due to a ceiling effect, as suggested by the authors.

The point of this study is that face-selective cells do give a weak response to bodies, but only when a body is presented below the receptive-field center. We do show responses to headless bodies in Figures 6 and 7. The experiment reported in Figure 6 (previously Figure 5) demonstrates that a body positioned below the RF can influence activity in the absence of an intact face regardless of what is within the RF – whether a face outline, face-like shape (noise), or even non-face object. The PSTHs comprise activity only within the face region above the body, not when the body is in the RF. The difference between each colored line in Figure 6 and the black line is sufficient to demonstrate the effect of a body being present below the RF. We now show responses to intact heads, face-shaped noise, face outlines, and objects with and without bodies (Figure 6—figure supplement 1). We also now show the response maps for each image in this experiment for both monkeys (Figure 6—figure supplements 1 and 2).

The question of whether these neurons would respond when a body is presented below a receptive field by itself is an interesting question. We already show those effects in Figure 7A. Additionally, we now show response maps for several images we tested during a pilot experiment that include headless mannequins and headless bodies (Figure 6—figure supplement 3). In each image, increased activity is apparent above each body.

4) The strong responses to the legs for the inverted body in Figure 6 is surprising to me. It clearly shows that the responses of these neurons to body stimuli without a head are more complex than just a response to an occluded face or absent head. The authors suggest that this is related to responses to short contours in the face cells. But if that is the case, can it not explain also the responses to the rectangular shapes (with also short contours) occluding the head in the images of Figure 3? Also, when it is related to short contours, why not then comparable strong responses to the images with a head (upper row of Figure 6)? It is clear to me that we do not understand the responses to these neurons. Also, how consistent was this leg response across the different sites? It also makes one wonder whether some of these "weird" responses are because the recordings are multi-unit, which will pool the responses of different neurons, some of which may not be face selective. In other words, it could be that the single neurons that respond to the legs are not face selective at all but mixed with face selective neurons in this patch. It would have been informative to show the responses of well-isolated single units in this case (and also in the other experiments).

For the experiment reported in Figure 7A, the responses are focused on and in-between the feet, but only when the feet are above a body. While this response was consistent across stimuli in this experiment, the ML arrays were generally unresponsive to feet in the other experiments. e.g., the responses around the feet are weaker than above the body in the bowling image in Figure 5, the monkey’s foot (except when photoshopped over the body) in Figure 6—figure supplements 1 and 2, the people carrying wood in Figure 4, the monkeys in Figure 4, the man carrying boxes in Figure 3, and the guy with the orangutan in Figure 1. As discussed in the results, we think the responses to the feet in Figure 7A are likely due to the peculiar situation that in the inverted body situation the feet are above a body. We showed in Figure 6 that pretty much any object above a body will elicit a response. The monkey object with body image in Figure 6—figure supplement 2 is a particularly clear example that whatever the base firing rate to a leg is, positioning it over the body increases its firing rate.

More broadly, the reviewer is concerned whether our effects could be the result of mixed selectivity from multi-unit data. We address the concern of mixed selectivity in the Essential Revisions section above. That said, we agree with the reviewer that the responses of these neurons are more complex than simply responding to an intact (or even occluded) face. We see robust activity to non-face objects – for example, we see hot spots on objects that have features similar to faces e.g., round, concentric circles or the tops of body-like structures (see Author response image 1). This does not diminish our findings. Our experiments were specifically aimed at probing the influence of a body below these neurons’ RFs. In particular the experiments reported in Figures 5, 6, and 7B target the effect of placing a body below the RF regardless of tuning properties to other visual features.

5) The authors suggest in the Discussion that the responses to the occluded head are not due merely to expectation. However, the latencies of the responses to the occluded head appear to be longer than to a real face (and, in some cases, the responses were (as for the non-face) preceded by inhibition (Figure 2, 3, 4 and 5 in M1)). Thus I feel one cannot rule out an expectation or top-down explanation of the effects.

An expectation or top-down account is hard to reconcile with our data. Stimuli were shown for only 100ms. Though latencies to occluded faces were longer than to an intact face, preferential responses to occluded faces were statistically significant by ~120-140ms, which seems too short for ‘top-down expectation’. Instead, we have now added to the discussion the idea that such ‘expectation’ could be broad enough to include the idea that the inputs to face cells become sculpted by experience to respond to things that are commonly seen with faces. Furthermore, the latency differences between PL and ML are also inconsistent with a strong topdown account. We now provide a new Figure 2 that demonstrates how the latencies vary in both PL and ML depending on the features present in the image, but PL always responds to intact of face features prior to ML. The fact that the eyeless faces have longer latencies (Figure 2) indicates these neurons likely have complex spatiotemporal RFs that are not driven by top-down expectation. We also now provide an analysis of face and body images using a hierarchical artificial network trained on image classification. We find that the representations of human faces and human bodies in deeper layers are more overlapping than the representations of faces and other categories (Figure 8). This network was not explicitly trained on faces, bodies or humans and thus demonstrates that representations of faces and bodies can merge over learning without any explicit training about faces and bodies.

To further explain our reasoning, we now provide the following Discussion:

“The idea that the context of a visual scene leads to the expectation, enhanced probability, or prior, of seeing some particular item is usually thought to involve top-down processes. However, if the feedforward inputs to, say, face-selective neurons, are sculpted by experience to be responsive not only to faces, but also to things frequently experienced in conjunction with faces, then such expectations would be manifest as the kind of complex response properties reported here.”

6) That the responses to the occluded heads result from experience makes sense but is speculation and not demonstrated in the present experiments. This should be made explicit.

We have softened our Discussion to make it clearer that our results are consistent with an interpretation that this contextual coding is dependent on experience without claiming that we have tested this. To bolster this hypothesis, we provide modeling results demonstrating how face and body representations can merge in a hierarchical system from statistical regularities of experience (even when not explicitly trained on those features).

[Editors' note: further revisions were suggested prior to acceptance, as described below.]

Reviewer #2:The authors performed an extensive revision, with new interesting data and modelling. They responded to all my concerns and overall I am happy with their revision. Nonetheless, I still have a couple of concerns that should be addressed by the authors.1) I understand that it is not possible to show the responses of all units to all stimuli, given the size of the data. I asked to show distribution of indices in order to get an idea about the across image and unit variability but the authors argue that this is not practical because of the variability in response latency amongst images and units. Such argument can always be made and then one should refrain from using indices (but the authors compute face and body indices). I think that when one uses net responses to compute indices, the variability in latency should not be a main cause of variance among indices of different neurons or images. One can employ an analysis window that captures the population response, computed across images and neurons, per experiment. The confidence intervals that are provided now depend on the number of observations; although they show the consistency of the mean population response, the CIs are not a straightforward metric of the variability among sites or images. Also, it is not clear whether the individual site examples shown in the figure supplements are representative of the observed diversity of response profiles. Hence, I still believe that distributions of indices will help to appreciate the variability among images and sites.

We now provide response histograms for both monkeys in Figures 3-6. We use the same time window as used for the response maps for each monkey. We show the distribution of responses across channels for individual images as well as the distribution of channel responses for the mean across images. Each graph shows that most channels show greater responses to the region above the body vs. comparison regions. The only condition we did not see this effect was for the face with body vs. face without body comparison in Figure 6 (leftmost graph), which, as previously discussed, did not show a significant difference in the population responses across images.

2) The authors provide novel data on the responses to images in which the eye is invisible. However, in the images shown in Figure 2, either more than the eyes are removed (top image) and it is unclear how these other changes affected the response to the disfigured face, or, the eyes are occluded by a visible occluder (bottom image). The latter is known to increase the response latency of IT neurons. To assess whether removing the eye from the image affects the response, it would have been better to present a face without eyes, the eye region being filled in with the neighboring texture, so that no additional changes (holes or bar occluder) are present. With the current images, I believe the authors should be more careful in interpreting the response and latency differences between the full face and the images in which the eyes were removed or occluded.

The reviewer raises a good point that in Figure 2, the images either had more than the eye removed or are presented with a visible occluder. We agree that the particular manner in which the eyes are removed from an image could affect neural responses. In Figure 2—figure supplement 1, we showed as the reviewer requested, an image in which the eye region was filled in with the neighboring texture (third image from the top). We took the face texture from sides of the monkey face and filled in the eye region. A similar shift in the latencies is evident in this image. We modified the main text to directly address this:

“A similar shift in response latencies was evident when replacing the eyes with a visible occluder, the background of the image, and surrounding texture of the face (Figure 2—figure supplement 1).”

3) The authors suggest that the responses to the occluded faces do not result from top-down processing, using as argument that the response latency differences between ML and PL remain for the occluded faces, which is a reasonable argument. However, PL and ML responses are compared directly only for the intact faces and the "invisible eye" images, but not for the face occlusion conditions. One way to show this is to compare in one figure the population PSTHs between ML and PL for the different conditions.

While we do not plot the responses to PL and ML on the same graph for face occlusion conditions, the difference in responses is apparent when comparing the graphs in Figures 3-5 to Figure 4—figure supplement 4 where responses occur prior to 100ms in PL and after 100ms in ML. We include as Author response image 3 a direct comparison of these PL and ML responses in monkey 1. Darker responses are ML and fainter responses are PL.

**Author response image 3. respfig3:** 

4) They write, when describing the novel and interesting Alexnet simulations, that "The Euclidean distances between image pairs within each category were smaller than between categories, indicating that images were more homogeneous within than between categories and highlighting that simple shape is a major component of these commonly tested "semantic" categories". However, the images also differ in texture and it is possible that not only shape but also texture is driving the category differences in Alexnet. In fact, some recent studies stressed the importance of texture cues for categorization in deep convolutional neural networks.

We agree with the reviewer that texture may also play an important role. We have modified the text to include reference to texture differences:

“The Euclidean distances between image pairs within each category were smaller than between categories, indicating that images were more homogenous within than between categories and highlighting that low-level features like shape, color, or texture are a major component of these commonly tested “semantic” categories (also see Dailey and Cottrell, 1999).”

5) The authors employ the Alexnet simulation to demonstrate that representations of faces and bodies can merge over learning. However, to show that learning is critical they should perform a similar analysis of the category representations in an untrained Alexnet, e.g. with random weights.

As requested, we now show that the similarity of faces and bodies in AlexNet is indeed specific to the trained network and not present in untrained networks. We simulated untrained networks by shuffling the weights of the trained AlexNet in each layer. This preserves the distribution of weights and allows for more direct comparisons with the trained network. We calculated image pair correlations on 500 permutations of the shuffled weights. In a new supplementary figure (Figure 8—figure supplement 1, we show that the mean and 97.5% percentiles of face-body correlations across permutations are below the face-body correlations in the trained network. Also, these correlations were no different than the correlations between faces and other categories (black vs. grey shaded regions in Figure 8—figure supplement 1).

6) The authors write "that convolutional networks can capture the representational structure of biological networks". It all depends on what is meant by "capture" here, but there is still a discrepancy between IT and classical feedforward deep net (like Alexnet) representations, e.g. see recent work by DiCarlo's group. I suggest to tone down this statement on the correspondence between deep nets and ventral stream representations.

We agree with the reviewer that convolutional nets such as AlexNet differ from the ventral stream in fundamental ways and it is erroneous to equate the two. We did not intend to draw such strong equivalencies. The second half of that quoted sentence highlights some of the ways in which these two differ. We agree with the reviewer that differences in representational structure also have been shown between deep nets and IT. We simply intended to convey that deep nets such as AlexNet can capture some of the representational structure such as these differences between image categories in order to demonstrate the effect of the statistics of experience on representations. This is consistent with DiCarlo’s (and others) work. As the reviewer suggested we have toned down this statement to avoid confusion:

“Though convolutional networks can capture some aspects of the representational structure of biological networks (Hong et al., 2016; Ponce et al., 2019),”

7) Can the authors explain why they permute the labels of the non-face images when computing the between face, non-face correlations (inset; Figure 8B); permutation of the non-face labels in this case should not make a difference, or perhaps I am missing something here.

Permuting all labels of non-face images including bodies and non-bodies tests whether correlations between faces and bodies are part of the same distribution of correlations for faces and other non-face, non-body images. By showing that the face-body correlations are stronger than 97.5% of permuted correlations, this indicates that the representations of faces and bodies are more overlapping than faces and other non-face, non-body images.

8) The authors should provide the explained variance of the 2D MDS configuration shown in their novel Figure 8A, so that the reader can appreciate how the distances plotted in 2D correspond to the original distances in the high-dimensional feature space.

We have added the variance explained to each of the 3 MDS plots. While the first 2 dimensions capture the majority of the variance for pixel intensity and relu7 (72% and 74%). The variance explained by the first two dimensions was substantially lower for relu1 (39%) and increased moving to deeper layers. We are using MDS as a visualization. Importantly, the correlation analyses reported in 8B are performed on the full dimensional feature spaces. We include as Author response image 4 the variance explained across all relu layers for 2D MDS.

**Author response image 4. respfig4:** 

9) The authors write in their Discussion "More broadly, these results highlight.… that the facilitation of face-selective neurons in the absence of a clear face reflects a lifetime of experience that bodies are usually accompanied by faces." As I noted before, there is no evidence yet that the contextual effects reported by the authors do result from experience during the lifetime. Please rephrase.

We have modified the text to more clearly indicate that we infer this:

“More broadly, these results highlight that neurons in inferotemporal cortex do not code objects and complex shapes in isolation. We propose that these neurons are sensitive to the statistical regularities of their cumulative experience and that the facilitation of face-selective neurons in the absence of a clear face reflects a lifetime of experience that bodies are usually accompanied by faces.”

10) Figure 1—figure supplement 1: the authors should indicate the t-values (with the corrected p = 0.05 threshold) of the face-object contrast in the fMRI maps.

We now include the t and p values for the fMRI maps:

“Data threshold at t(2480) > 3.91 (p < 0.0001 FDR-corrected) and t(1460) > 3.91 (p < 0.0001 FDR-corrected) for monkeys 1 and 2, respectively.”

11) Figure 1—figure supplement 3: can the authors explain the above-baseline responses in M1, ML before 50 ms after stimulus onset – one would expect that soon after stimulus onset responses equal to baseline because of the response latency of the neurons.

We thank the reviewer for catching this. The responses were baseline-subtracted, but the axis labels were incorrect. We’ve fixed the graph to correct this. We confirm that the other response graphs were not incorrectly labeled.